# Establishing Linear Surrogate Regret Bounds for Convex Smooth Losses via Convolutional Fenchel–Young Losses

Yuzhou Cao[1]   Han Bao[2]   Lei Feng[3*]   Bo An[1,4]

[1] College of Computing and Data Science, Nanyang Technological University
[2] The Institute of Statistical Mathematics
[3] School of Computer Science and Engineering, Southeast University
[4] Skywork AI
yuzhou002@e.ntu.edu.sg   bao.han@ism.ac.jp
fenglei@seu.edu.cn   boan@ntu.edu.sg

## Abstract

Surrogate regret bounds, also known as excess risk bounds, bridge the gap between the convergence rates of surrogate and target losses. The regret transfer is lossless if the surrogate regret bound is linear. While convex smooth surrogate losses are appealing in particular due to the efficient estimation and optimization, the existence of a trade-off between the loss smoothness and linear regret bound has been believed in the community. Under this scenario, the better optimization and estimation properties of convex smooth surrogate losses may inevitably deteriorate after undergoing the regret transfer onto a target loss. We overcome this dilemma for arbitrary discrete target losses by constructing a convex smooth surrogate loss, which entails a linear surrogate regret bound composed with a tailored prediction link. The construction is based on Fenchel–Young losses generated by the *convolutional negentropy*, which are equivalent to the infimal convolution of a generalized negentropy and the target Bayes risk. Consequently, the infimal convolution enables us to derive a smooth loss while maintaining the surrogate regret bound linear. We additionally benefit from the infimal convolution to have a consistent estimator of the underlying class probability. Our results are overall a novel demonstration of how convex analysis penetrates into optimization and statistical efficiency in risk minimization.

## 1 Introduction

The risk of a machine learning model is often measured by the expectation of a target loss $\ell$ that quantifies the error between the model prediction $t$ and a natural label $y$. However, minimizing the target risk over a dataset is often computationally hard because a target prediction problem is usually discretely structured, including multiclass, multilabel, top-$k$ prediction problems. For this reason, a tractable surrogate risk induced by a *surrogate loss* $L(\boldsymbol{\theta}, y)$ serves as an essential proxy with a score $\boldsymbol{\theta} \in \mathbb{R}^d$. The resulting surrogate optimization is no longer discretely constrained.

An ideal surrogate loss should be convex, smooth (or entailing a Lipschitz continuous gradient), and calibrated toward a given target prediction problem. Convexity and smoothness have been fundamental both in optimization and statistical estimation—indeed, classical optimization theory reveals that convex smooth functions can be optimized with first-order methods more efficiently than

---

*Corresponding author.

39th Conference on Neural Information Processing Systems (NeurIPS 2025).

non-smooth functions [67, 83]. In addition, several studies have demonstrated that the convexity and smoothness can further enhance fast rates in the risk estimation [87, 98]. Meanwhile, calibration is regarded as a minimal requirement on surrogate losses, ensuring that the surrogate risk minimization leads to the target risk minimization [12, 88].[2] To establish a relationship between surrogate and target risks, *surrogate regret bounds* play a crucial role. Therein the regret (or the suboptimality) of a target loss is controlled by that of a surrogate loss through a non-decreasing rate function $\psi : \mathbb{R}_{\geq 0} \to \mathbb{R}_{\geq 0}$. A surrogate regret bound can be informally written as follows: for any score vector $\boldsymbol{\theta} \in \mathbb{R}^d$ and a class distribution $\boldsymbol{\eta}$,

$$\texttt{Regret}_\ell(\varphi(\boldsymbol{\theta}), \boldsymbol{\eta}) \leq \psi\left(\texttt{Regret}_L(\boldsymbol{\theta}, \boldsymbol{\eta})\right), \tag{1}$$

where $\varphi$ is a prediction link that converts a score vector $\boldsymbol{\theta}$ into a target prediction $\varphi(\boldsymbol{\theta})$. If the regret rate function is linear $\psi(r) = \mathcal{O}(r)$, which is the best possible (data-independent) regret rate,[3] then the optimization and estimation errors of the surrogate loss are optimally translated to the target loss. For example, under the binary classification target loss, Bartlett et al. [12] demonstrated that the linear regret rate is possible with the hinge loss. Later, symmetric losses [24] and polyhedral losses [39] were shown to yield the linear regret rate. Unfortunately, these loss functions lack either convexity or smoothness, which deteriorates the optimization and estimation errors of the surrogate regret, even if they enjoy linear regret bounds. By contrast, a square-root regret rate $\psi(r) = \mathcal{O}(\sqrt{r})$ is common for convex smooth surrogate losses, such as the logistic, exponential, and squared losses [12, 69, 39]. These losses typically enjoy better optimization and estimation properties, yet suffer from larger target regrets due to the suboptimal regret rate. Therefore, it has been open to develop a convex smooth surrogate loss without sacrificing the linear regret rate.

Notwithstanding, the previous literature in this line has implied that such an ideal surrogate loss may be inconceivable. Mahdavi et al. [50] considered this question with an interpolated loss between the hinge (non-smooth) and logistic (smooth) losses and showed that we inevitably face the trade-off between generalization and optimization unless strong distributional assumptions are imposed. Further, Frongillo and Waggoner [39] proved that locally smooth and strongly convex losses must suffer from a square-root regret rate at least. Ramaswamy et al. [76] blame convex smooth losses for redundantly modeling continuous class distributions, which is unnecessary if our goal is merely to solve discrete target problems.

In this paper, we demonstrate that this seemingly impossible trade-off can be overcome for *arbitrary* discrete target losses. Specifically, we build a convex smooth surrogate loss built upon the framework of Fenchel–Young losses [16]—a framework to generate a loss function from a generalized negentropy and its conjugate. In a nutshell, our main results are summarized as follows:

**Theorem 1 (Informal version of Theorem 15)** *For any discrete target loss $\ell$, there exist a convex smooth surrogate loss $L$ (defined over a score $\boldsymbol{\theta} \in \mathbb{R}^d$) and prediction link $\varphi$ such that the surrogate regret bound (1) holds with some linear rate $\psi(r) = \mathcal{O}(r)$.*

This existence is proved constructively, which provides a systematic framework on the construction of convex smooth surrogate losses and their corresponding prediction links with linear surrogate regret bounds. Our high-level construction, which significantly leverages convex analysis and is detailed in Section 3.1, proceeds as follows. First, a user chooses a strongly convex negentropy with some regularity conditions, such as the Shannon negentropy. We encode the structure of a target loss $\ell$ into the chosen base negentropy by adding the negative Bayes risk of $\ell$. This additivity eventually translates into the infimal convolution in the induced Fenchel–Young loss, which we call the *convolutional* Fenchel–Young loss. The convolutional Fenchel–Young loss is endowed with the convexity and smoothness arising from the base negentropy, shown in Section 3.2. Then we can obtain a prediction link via the infimal convolution. Paired with the convolutional Fenchel–Young loss, it admits a linear surrogate regret bound, demonstrated in Section 3.3.

In addition, we improve the multiplicative term in the initial linear surrogate regret bound in Section 3.4 to make the bound tighter, which exploits the low-rank structure of a target loss $\ell$. As a by-product of the loss smoothness, we can provide a Fisher-consistent probability estimator of the underlying probability in Section 3.5. Finally, Section 4 instantiates the framework of the convolutional Fenchel–Young loss for the multiclass classification problem, highlighting the efficient computation

---

[2]Readers should distinguish this calibration property from calibrated prediction [38].

[3]A super-linear bound is possible with distribution-dependent losses [97], which we do not consider here.

of the prediction link. More examples of target prediction problems, such as classification with rejection and multilabel ranking, are available in Appendix D.

## 1.1 Related Works

**Surrogate losses for general discrete prediction problems.** A discrete prediction problem aims to predict $t$ that minimizes a discrete target loss $\ell(t, y)$ over the class distribution, and the study of convex surrogates for $\ell$ is of significant interest. Extensive research has been conducted on surrogate losses for specific discrete tasks, including but not limited to classification [12, 101, 89, 93, 85, 66, 64, 81], top-$k$ [95, 90], and multilabel learning [40, 100, 54, 49, 65].

In contrast to *ad-hoc* approaches, recent studies have advanced the principled design of calibrated convex surrogate losses for general discrete prediction problems, without imposing restrictions on the target losses. In Ramaswamy and Agarwal [73] and Ramaswamy et al. [75], surrogate losses based on the squared loss and error correcting output codes are proposed, respectively, which embed the structure of a target problem into a surrogate loss and are shown to be calibrated with the corresponding target losses. The design of calibrated surrogate losses has also been extensively studied in the context of structured prediction [29, 74, 26, 27, 70, 68, 69, 23], where the structural/low-rank properties of target losses are exploited to construct more efficient surrogates. As opposed to the smooth losses utilized in the works above, polyhedral losses [34, 35, 37] have attracted attention recently, which provides a systematic framework for efficiently constructing calibrated yet non-smooth convex losses based on a discrete target loss.

**Surrogate regret bounds.** Surrogate regret bounds have been well studied for margin losses under binary and multiclass classification [101, 12, 89, 84, 56, 86], where the target loss is the 0-1 loss. Similarly, proper losses [21, 42, 78, 3, 45, 63] have been shown to provide surrogate regret bounds *w.r.t.* the $L_1$ distance between the estimated and true class probabilities [77, 8, 9], which facilitates analyses for downstream tasks. For structured prediction, surrogate regret bounds are sometimes called comparison inequalities [26, 70, 68, 14], which are typically of square-root type. Notably, Frongillo and Waggoner [39] demonstrated that polyhedral losses exhibit linear surrogate regret bounds for general discrete prediction problems, covering various piecewise linear non-smooth losses [95, 76] as special cases. However, they lack the smoothness. To the contrary, Mao et al. [53] and Mao et al. [52] propose smooth losses with linear regret rates but lacking convexity.

Whereas a growing line of research has focused on $\mathcal{H}$-consistency to analyze how the restriction to a hypothesis space $\mathcal{H}$ affects consistency [48, 99, 7], we implicitly suppose that the hypothesis space is all measurable functions because the analysis is often more transparent and it is reasonable to suppose that the hypothesis space is sufficiently expressive under the overparametrization regime. The extension to $\mathcal{H}$-consistency is rather straightforward by integrating the minimizability gap [52, 55, 54, 57, 51].[4]

## 2 Preliminaries

Let $[d] := \{1, 2, \cdots, d\}$ and $[\![\cdot]\!]$ is the Iverson bracket. The $p$-norm is denoted by $\| \cdot \|_p$, which we assume to be the 2-norm unless otherwise noted. Let $\overline{\mathbb{R}} := \mathbb{R} \cup \{\infty\}$ be the extended real-line and $\Delta^d := \{\boldsymbol{\eta} \in \mathbb{R}^d_{\geq 0} : \|\boldsymbol{\eta}\|_1 = 1\}$ the $d$-simplex. For a set $\mathcal{S} \subseteq \mathbb{R}^d$, $\mathrm{int}(\mathcal{S})$ and $\mathrm{relint}(\mathcal{S})$ are its interior and relative interior, respectively, and $\mathrm{conv}(\mathcal{S})$ is its convex hull. The indicator function is denoted by $\mathbb{I}_{\mathcal{S}} : \mathbb{R}^d \to \{0, +\infty\}$, where $\mathbb{I}_{\mathcal{S}}(\boldsymbol{\theta}) = 0$ if $\boldsymbol{\theta} \in \mathcal{S}$ and $+\infty$ otherwise. For a function $f : \mathbb{R}^d \to \overline{\mathbb{R}}$, $\mathrm{dom}(f) := \{\boldsymbol{\theta} \in \mathbb{R}^d : f(\boldsymbol{\theta}) < +\infty\}$ is its effective domain. A function $f$ is extended to be set-valued with slight abuse of notation by $f(\mathcal{S}) := \{f(s) : s \in \mathcal{S}\}$. The Fenchel conjugate of $\Omega$ is $\Omega^*(\boldsymbol{\theta}) := \sup_{\boldsymbol{p} \in \mathrm{dom}(\Omega)}\{\boldsymbol{\theta}^\top \boldsymbol{p} - \Omega(\boldsymbol{p})\}$. The identity matrix is $I$. The canonical basis of the Euclidean space is denoted by $\{\boldsymbol{e}_i\}$, where the dimensionality depends on the contexts.

### 2.1 Discrete Prediction Problems and Target Losses

Let $\mathcal{Y} = [K]$ be the finite class space. The class distribution on $\mathcal{Y}$ is $\boldsymbol{\eta} \in \Delta^K$ such that class $Y = y$ has probability $\eta_y$. A discrete prediction problem aims to find a target prediction $t$ from the finite

---

[4]This condition can be relaxed to the *well-specified hypothesis space*, under which the hypothesis space is required to contain at least one population risk minimizer, rather than the entire space of measurable functions.

prediction space $\widehat{\mathcal{Y}} := [N]$ for each $\boldsymbol{\eta}$ by minimizing a discrete *target loss* $\ell : \widehat{\mathcal{Y}} \times \mathcal{Y} \to \mathbb{R}$ over $y \sim \boldsymbol{\eta}$. The averaged target loss is called the target risk: $R_\ell(t, \boldsymbol{\eta}) := \mathbb{E}_{y \sim \boldsymbol{\eta}}[\ell(t, y)] = \langle \boldsymbol{\eta}, \boldsymbol{\ell}(t) \rangle$, where $\boldsymbol{\ell}(t) \in \mathbb{R}^K$ is the loss vector such that $\boldsymbol{\ell}(t)_y = \ell(t, y)$ for each $y \in \mathcal{Y}$. The Bayes risk of $\ell$ at $\boldsymbol{\eta}$ is $\underline{R}_\ell(\boldsymbol{\eta}) := \min_{t \in \widehat{\mathcal{Y}}} R_\ell(t, \boldsymbol{\eta})$, which is a concave function of $\boldsymbol{\eta}$. The suboptimality of a prediction $t$ *w.r.t.* the target loss $\ell$ over a class distribution $\boldsymbol{\eta}$ is characterized by the *target regret*, which is the gap between its risk and the Bayes risk.

**Definition 2 (Target regret)** *Given a discrete target loss $\ell$, the target regret of a prediction $t$ w.r.t. a class probability $\boldsymbol{\eta} \in \Delta^K$ is defined as follows:*

$$\texttt{Regret}_\ell(t, \boldsymbol{\eta}) := R_\ell(t, \boldsymbol{\eta}) - \underline{R}_\ell(\boldsymbol{\eta}). \tag{2}$$

**Equivalent lower-dimensional decomposition.** By encoding every possible class label $y \in \mathcal{Y}$ into the canonical basis $\boldsymbol{e}_y \in \mathbb{R}^K$, we can express any target loss in the following form:

$$\ell(t, y) = \langle \boldsymbol{e}_y, \boldsymbol{\ell}(t) \rangle. \tag{3}$$

An equivalent but more efficient decomposition of (3), known as Structure Encoding Loss Functions (SELF), was introduced in Ciliberto et al. [26], which allows lower-dimensional label encodings. It has been widely used to construct efficient loss functions, and further generalized via Affine Decomposition in Blondel [14, (12)]. We also adopt a general form to represent discrete target losses.

**Definition 3 (($\boldsymbol{\rho}, \boldsymbol{\ell}^\rho$)-decomposition)** *For a discrete target loss $\ell : \widehat{\mathcal{Y}} \times \mathcal{Y} \to \mathbb{R}$, its $(\boldsymbol{\rho}, \boldsymbol{\ell}^\rho)$-decomposition is given as follows:*

$$\ell(t, y) = \langle \boldsymbol{\rho}(y), \boldsymbol{\ell}^\rho(t) \rangle + c(y), \tag{4}$$

*where $\boldsymbol{\rho} : \mathcal{Y} \to \mathbb{R}^d$ is a label encoding function that maps discrete labels into the $d$-dimensional Euclidean space, $\boldsymbol{\ell}^\rho : \widehat{\mathcal{Y}} \to \mathbb{R}^d$ is the corresponding loss encoding function, and $c : \mathcal{Y} \to \mathbb{R}$ is the remainder independent of prediction $t$.*

This loss decomposition contains the Affine Decomposition [14, (12)]. Any discrete target loss admits such a decomposition via the trivial choice $\boldsymbol{\rho}(y) = \boldsymbol{e}_y$, $\boldsymbol{\ell}^\rho(t) = \boldsymbol{\ell}(t)$, and $c = 0$ with $d = K$, which immediately recovers (3). With a properly chosen $\boldsymbol{\rho}$, the encoded cardinality $d$ can often be greatly smaller than $K$, particularly in the context of structured prediction. For example, consider multilabel classification, where $\mathcal{Y} = \widehat{\mathcal{Y}} = [K]$, $K = 2^d$, and each $y \in [K]$ corresponds to a unique binary vector $\boldsymbol{\nu}(y) \in \{0, 1\}^d$ representing a set of binary labels. Suppose our target loss is the Hamming loss $\ell_H(t, y) := \mathbf{1}^\top (\boldsymbol{\nu}(t) + \boldsymbol{\nu}(y)) - 2\langle \boldsymbol{\nu}(t), \boldsymbol{\nu}(y) \rangle$, which is the Hamming distance between $\boldsymbol{\nu}(t)$ and $\boldsymbol{\nu}(y)$. This entails the decomposition $\boldsymbol{\rho}(y) = \mathbf{1} - 2\boldsymbol{\nu}(y)$, $\boldsymbol{\ell}^\rho(t) = \boldsymbol{\nu}(t)$, and $c(y) = \mathbf{1}^\top \boldsymbol{\nu}(y)$. Here, $d = \log_2 K$ significantly reduces the dimensionality from the cardinality of $\mathcal{Y}$. We refer reader to Blondel [14, Appendix A] and Appendix D for more examples of discrete prediction problems. The cardinality $d$ of label encoding $\boldsymbol{\rho}$ is closely related to surrogate regret bounds later in Section 3.4.

## 2.2 Surrogate Losses and Surrogate Regret Bounds

Unlike discrete target losses assessing a discrete prediction $t \in \widehat{\mathcal{Y}}$, a surrogate loss $L : \mathbb{R}^d \times \mathcal{Y} \to \overline{\mathbb{R}}$ receives a continuous score $\boldsymbol{\theta} \in \mathbb{R}^d$. Then a prediction link $\varphi : \mathbb{R}^d \to \widehat{\mathcal{Y}}$ is used to transform a score $\boldsymbol{\theta}$ to a prediction $t$. Its risk and Bayes risk are defined as $R_L(\boldsymbol{\theta}, \boldsymbol{\eta}) := \mathbb{E}_{y \sim \boldsymbol{\eta}}[L(\boldsymbol{\theta}, y)] = \langle \boldsymbol{\eta}, \boldsymbol{L}(\boldsymbol{\theta}) \rangle$ and $\underline{R}_L(\boldsymbol{\eta}) := \inf_{\boldsymbol{\theta} \in \mathbb{R}^d} R_L(\boldsymbol{\theta}, \boldsymbol{\eta})$, respectively, where $\boldsymbol{L}(\boldsymbol{\theta}) := [L(\boldsymbol{\theta}, y)]_{y=1}^K$. The regret of a surrogate loss is defined as the gap between the risk of a score $\boldsymbol{\theta}$ and the Bayes risk, similar to the target regret.

**Definition 4 (Surrogate regret)** *Given a surrogate loss $L$, the surrogate regret of score $\boldsymbol{\theta}$ w.r.t. a class probability $\boldsymbol{\eta} \in \Delta^K$ is defined as follows:*

$$\texttt{Regret}_L(\boldsymbol{\theta}, \boldsymbol{\eta}) := R_L(\boldsymbol{\theta}, \boldsymbol{\eta}) - \underline{R}_L(\boldsymbol{\eta}). \tag{5}$$

For a desirable surrogate loss, the convergence of its surrogate regret will dominate the target regret. That is, for a fixed class distribution $\boldsymbol{\eta}$, a convergent $\boldsymbol{\theta}$ such that $\texttt{Regret}_L(\boldsymbol{\theta}, \boldsymbol{\eta}) \to 0$ should imply $\texttt{Regret}_\ell(\varphi(\boldsymbol{\theta}), \boldsymbol{\eta}) \to 0$. This convergence relationship indicates that the target regret minimization can be achieved by the minimization of the surrogate regret adopted with an appropriate prediction link. A surrogate regret bound offers a quantitative characterization of this relationship through a regret rate function $\psi$, which is the key focus of this work.

**Definition 5 (Surrogate regret bound)** *A surrogate loss $L$ and prediction link $\varphi$ entail a surrogate regret bound w.r.t. target $\ell$ with a non-decreasing regret rate function $\psi : \mathbb{R}_{\geq 0} \to \mathbb{R}_{\geq 0}$ satisfying $\psi(0) = 0$ if the following inequality holds:*

$$\text{Regret}_\ell(\varphi(\boldsymbol{\theta}), \boldsymbol{\eta}) \leq \psi(\text{Regret}_L(\boldsymbol{\theta}, \boldsymbol{\eta})), \quad \text{for any } (\boldsymbol{\theta}, \boldsymbol{\eta}) \in \mathbb{R}^d \times \Delta^K. \tag{6}$$

While a linear regret rate $\psi(r) = \mathcal{O}(r)$ is the best possible, previously know linear-rate losses are either non-smooth or non-convex, including the (non-smooth) hinge loss [12] and (non-convex) sigmoid loss [24]. These loss functions may face challenges in optimization and estimation [87]. In this work, we aim to develop a framework that facilitates the use of convex smooth surrogates with linear surrogate regret bounds.

### 2.3 Fenchel–Young Loss

In this work, we build upon Fenchel–Young losses [16, 18, 59] to construct convex smooth surrogates equipped with linear regret rate functions. Let us review Fenchel–Young losses first.

**Definition 6 (Fenchel–Young loss)** *For $\Omega : \mathbb{R}^d \to \overline{\mathbb{R}}$, the associated Fenchel–Young loss $L_\Omega : \text{dom}(\Omega^*) \times \text{dom}(\Omega) \to \mathbb{R}_{\geq 0}$ is defined as follows:*

$$L_\Omega(\boldsymbol{\theta}, \boldsymbol{p}) = \Omega(\boldsymbol{p}) + \Omega^*(\boldsymbol{\theta}) - \langle \boldsymbol{\theta}, \boldsymbol{p} \rangle. \tag{7}$$

Similar constructions can also be found in Duchi et al. [33], Agarwal et al. [2], which focus on multiclass classification. The function $\Omega$ is often regarded as a generalized negentropy, which equals the negative of the Bayes risk of its induced Fenchel–Young loss. Fenchel–Young losses often impose additional requirements on the domains of $\Omega$ and $\Omega^*$, tailored to specific target problems, and we will highlight them when necessary.

Fenchel–Young losses possess favorable properties: they are inherently convex (in score $\boldsymbol{\theta}$) by definition. Moreover, we can use the map $\nabla\Omega^*(\boldsymbol{\theta})$ to obtain the mean $\mathbb{E}_{y\sim\boldsymbol{\eta}}[\boldsymbol{\rho}(y)]$, which is the class distribution $\boldsymbol{\eta}$ under multiclass classification with $\boldsymbol{\rho}(y) = \boldsymbol{e}_y$; or linear properties in general [1]. Many common losses are encompassed in Fenchel–Young losses, including the cross-entropy loss, squared loss, and Crammer–Singer loss [31]. For example, $\Omega(\boldsymbol{p}) = \frac{1}{2}\|\boldsymbol{p}\|^2 + \mathbb{I}_{\Delta^K}(\boldsymbol{p})$ generates the sparsemax loss, which induces the sparsemax as a Fisher-consistent estimator of $\boldsymbol{\eta}$ [58].

**Connections to information geometry.** While Fenchel–Young losses are systematically defined under the convex-analytic formulation, it also has a longstanding history in information geometry, particularly through its connection to the Bregman divergence and related generalizations [72]. In Blondel et al. [16], it is noted that a Fenchel–Young loss can be interpreted as the mixed-type Bregman divergence [4]. When $\Omega$ is of Legendre-type, $\Omega$ can yield the dual coordinate system and the Fenchel–Young loss is further equivalent to the *canonical divergence* generated by $\Omega$ [5, Eq. (3.44)].

## 3 Convex Smooth Surrogates with Linear Surrogate Regret Bounds

In this section, we first show our construction of surrogate losses and prediction links. After showing that this loss is convex and smooth with an appropriately chosen base negentropy (Theorem 11), we move on to the main result of this paper, linear surrogate regret bounds with convex smooth surrogate losses (Theorem 13). We further discuss the computational aspects and improve the regret bound constant. The missing proofs are deferred to Appendix B.

### 3.1 Construction

We design a convex smooth surrogate loss built upon the framework of Fenchel–Young losses. To make its surrogate regret bound linear, we craft a negentropy by delicately leveraging the structure of a target prediction problem. Denote by $T$ the polyhedral convex function $T(\boldsymbol{p}) = -\min_{t\in\widehat{\mathcal{Y}}} \langle \boldsymbol{p}, \boldsymbol{\ell}^{\boldsymbol{\rho}}(t) \rangle$. Then, we have the following definition.

**Definition 7 (Convolutional negentropy)** *Suppose $(\boldsymbol{\rho}, \boldsymbol{\ell}^{\boldsymbol{\rho}})$-decomposition for a target loss $\ell$. For a negentropy $\Omega : \mathbb{R}^d \to \overline{\mathbb{R}}$, its convolutional negentropy is defined as follows:*

$$\Omega_T(\boldsymbol{p}) = \Omega(\boldsymbol{p}) + T(\boldsymbol{p}) \tag{8}$$

While commonly used negentropy in Fenchel–Young losses, such as the Shannon negentropy and norm negentropy [19], are unstructured toward a target loss, the convolutional negentropy $\Omega_T$ encodes a target loss $\ell$ explicitly into the base negentropy $\Omega$ via $T$. This polyhedral convex function $T$ is an affinely transformed negative Bayes risk of a target loss $\ell$ when $\boldsymbol{p} \in \text{conv}\{\boldsymbol{\rho}(\mathcal{Y})\}$. Indeed, with the linear property $\boldsymbol{p} = \mathbb{E}_{y \sim \boldsymbol{\eta}}[\boldsymbol{\rho}(y)]$, i.e., expectation of $\boldsymbol{\rho}(y)$ w.r.t. class probability $\boldsymbol{\eta}$:

$$T(\boldsymbol{p}) = -\min_{t \in \widehat{\mathcal{Y}}} \langle \boldsymbol{p}, \boldsymbol{\ell}^{\boldsymbol{\rho}}(t) \rangle = -\min_{t \in \widehat{\mathcal{Y}}} \mathbb{E}_{y \sim \boldsymbol{\eta}}[\langle \boldsymbol{\rho}(y), \boldsymbol{\ell}^{\boldsymbol{\rho}}(t) \rangle] = \sum_{y \in [K]} \eta_y c(y) - \underline{R}_\ell(\boldsymbol{\eta}). \quad (9)$$

The base negentropy $\Omega$ is up to our choice. Before deriving the conjugate of $\Omega_T$, we make the following requirement on $\Omega$ throughout this work for a well-behaved convolutional negentropy.

**Condition 1** $\Omega$ *is proper convex and lower-semicontinuous (l.s.c.), and satisfies* $\text{conv}(\boldsymbol{\rho}(\mathcal{Y})) \subseteq \text{dom}(\Omega)$ *and* $\text{dom}(\Omega^*) = \mathbb{R}^d$.

It is a mild requirement. Indeed, proper convexity and lower-semicontinuity merely aim to avoid pathologies, and the domain assumptions are met by a differentiable and finite $\Omega$ over the valid prediction space $\text{conv}(\boldsymbol{\rho}(\mathcal{Y}))$. The assumption on $\Omega^*$ is crucial for the induced loss to have unconstrained domain $\mathbb{R}^d$. Then we show an explicit form of the conjugated convolutional negentropy.

**Lemma 8 (Conjugate of $\Omega_T$)** *Suppose Condition 1 holds, then $\Omega_T^*$ is proper convex and l.s.c. with* $\text{dom}(\Omega_T^*) = \mathbb{R}^d$*, and can be expressed as follows:*

$$\Omega_T^*(\boldsymbol{\theta}) = \inf_{\boldsymbol{\pi} \in \Delta^N} \Omega^*(\boldsymbol{\theta} + \mathcal{L}^{\boldsymbol{\rho}} \boldsymbol{\pi}) \quad \text{for any } \boldsymbol{\theta} \in \mathbb{R}^d, \quad (10)$$

*where $\mathcal{L}^{\boldsymbol{\rho}} \in \mathbb{R}^{d \times N}$ is the loss matrix with the $t$-th column being $\boldsymbol{\ell}^{\boldsymbol{\rho}}(t) \in \mathbb{R}^d$ for $t \in \widehat{\mathcal{Y}}$. In addition, there always exists $\boldsymbol{\pi} \in \Delta^N$ that achieves the infimum of (10) for any $\boldsymbol{\theta} \in \mathbb{R}^d$.*

The conjugated form (10) owes to the *infimal convolution* between the base negentropy $\Omega$ and the target Bayes risk $T$. We coined the name of the convolutional negentropy inspired by this structure. Intuitively, the conjugated convolutional negentropy $\Omega_T^*$ can be viewed as a "perturbed" version of the conjugated base negentropy $\Omega^*$ toward the direction of the target loss matrix $\mathcal{L}^{\boldsymbol{\rho}}$. This result facilitates a more concrete formulation of the Fenchel–Young loss generated by $\Omega_T$, which we refer to as the convolutional Fenchel–Young loss—the central focus of this work.

**Definition 9 (Convolutional Fenchel–Young loss)** *Suppose that a discrete target loss $\ell$ enjoys $(\boldsymbol{\rho}, \boldsymbol{\ell}^{\boldsymbol{\rho}})$-decomposition. For a negentropy $\Omega : \mathbb{R}^d \to \overline{\mathbb{R}}$ satisfying Condition 1, the convolutional Fenchel–Young loss $L_{\Omega_T} : \mathbb{R}^d \times \mathcal{Y} \to \overline{\mathbb{R}}$ induced by the convolutional negentropy $\Omega_T$ is defined as follows:*

$$L_{\Omega_T}(\boldsymbol{\theta}, y) = \min_{\boldsymbol{\pi} \in \Delta^N} \Omega^*(\boldsymbol{\theta} + \mathcal{L}^{\boldsymbol{\rho}} \boldsymbol{\pi}) + \Omega_T(\boldsymbol{\rho}(y)) - \langle \boldsymbol{\theta}, \boldsymbol{\rho}(y) \rangle. \quad (11)$$

Note that $L_{\Omega_T}(\boldsymbol{\theta}, \cdot)$ is deliberately constrained to the class space $\mathcal{Y}$ instead of the general domain $\text{dom}(\Omega_T)$, to align with the surrogate loss form introduced in Section 2.2.

Given a surrogate loss, we need to specify a prediction link. For general discrete prediction problems where $\mathcal{Y} \neq \widehat{\mathcal{Y}}$, a prediction link yields a discrete prediction in $\widehat{\mathcal{Y}}$ from a score $\boldsymbol{\theta} \in \mathbb{R}^d$ obtained through minimizing the surrogate loss. Thus, a loss and link are the two sides of the same coin. Unlike the standard argmax link in multiclass classification [89, 94], we create an alternative argmax-like prediction link based on the minimizer of (10).

**Definition 10 ($\boldsymbol{\pi}$-argmax link)** *Let $\Pi : \mathbb{R}^d \to 2^{\Delta^N}$ be a set-valued map defined as follows:*

$$\Pi(\boldsymbol{\theta}) := \text{argmin}\left\{\Omega^*(\boldsymbol{\theta} + \mathcal{L}^{\boldsymbol{\rho}} \boldsymbol{\pi}) : \boldsymbol{\pi} \in \Delta^N\right\} \quad \text{for any } \boldsymbol{\theta} \in \mathbb{R}^d. \quad (12)$$

*Let $\boldsymbol{\pi} : \mathbb{R}^d \to \Delta^N$ be a selector of $\Pi$ such that $\boldsymbol{\pi}(\boldsymbol{\theta}) \in \Pi(\boldsymbol{\theta})$ for all $\boldsymbol{\theta} \in \mathbb{R}^d$.[5] Then the $\boldsymbol{\pi}$-argmax link $\varphi : \mathbb{R}^d \to \widehat{\mathcal{Y}}$ is defined as*

$$\varphi(\boldsymbol{\theta}) \in \text{argmax}\left\{\pi_t(\boldsymbol{\theta}) : t \in \widehat{\mathcal{Y}}\right\} \quad \text{for any } \boldsymbol{\theta} \in \mathbb{R}^d,$$

*where the tie can be broken arbitrarily.*

---

[5]We slightly abuse the notation $\boldsymbol{\pi}$ by using it for both a vector and a vector-valued mapping; in particular, the $\boldsymbol{\pi}$ appearing in the $\boldsymbol{\pi}$-argmax link refers to the selector function.

The existence of such links is guaranteed as long as $\Pi(\boldsymbol{\theta})$ is non-empty for every $\boldsymbol{\theta} \in \mathbb{R}^d$, which is guaranteed by Lemma 8. While the standard argmax link returns a prediction $t \in \widehat{\mathcal{Y}}$ with the maximum score $\theta_t$, the $\boldsymbol{\pi}$-argmax link returns $t$ with the maximum $\pi_t$. This probabilistic quantity $\boldsymbol{\pi} \in \Pi(\boldsymbol{\theta})$ defined via (10) distorts the original score $\boldsymbol{\theta}$ by leveraging the target loss structure $\mathcal{L}^{\boldsymbol{\rho}}$.

## 3.2 Convexity and Smoothness

Before discussing surrogate regret bounds, we verify that convolutional Fenchel–Young losses are indeed convex and smooth, implied by the conjugacy between smoothness and strong convexity [44].

**Corollary 11 (Convexity and smoothness of $L_{\Omega_T}$)** *Suppose that a discrete target loss $\ell$ enjoys $(\boldsymbol{\rho}, \ell^{\boldsymbol{\rho}})$-decomposition. For a base negentropy $\Omega$, we additionally suppose that Condition 1 is satisfied. If $\Omega$ is strictly convex on $\mathrm{dom}(\Omega)$, $L_{\Omega_T}(\cdot, y)$ is convex and differentiable over $\mathbb{R}^d$ for any $y \in \mathcal{Y}$. If $\Omega$ is additionally strongly convex on $\mathrm{dom}(\Omega)$, $L_{\Omega_T}(\cdot, y)$ is smooth over $\mathbb{R}^d$ for any $y \in \mathcal{Y}$.*

There are several exemplar base negentropies fulfilling the conditions of Corollary 11. For example, the squared norm $\Omega(\boldsymbol{p}) = \frac{1}{2}\|\boldsymbol{p}\|^2$ is strongly convex with the self-conjugate $\Omega^*(\boldsymbol{\theta}) = \frac{1}{2}\|\boldsymbol{\theta}\|^2$. In this case, $\Omega$ is effective over the entire $\mathbb{R}^d$ and hence includes $\mathrm{conv}(\boldsymbol{\rho}(\mathcal{Y}))$, satisfying Condition 1. In light of the target-loss decomposition (3), we have $\mathrm{conv}(\boldsymbol{\rho}(\mathcal{Y})) = \Delta^K$. In this case, the Shannon negentropy $\Omega(\boldsymbol{p}) = \langle \boldsymbol{p}, \ln \boldsymbol{p} \rangle + \mathbb{I}_{\Delta^K}(\boldsymbol{p})$ is strongly convex on $\Delta^K$ with its conjugate $\Omega^*(\boldsymbol{\theta}) = \ln\langle \exp(\boldsymbol{\theta}), \boldsymbol{1} \rangle$ satisfying $\mathrm{dom}(\Omega^*) = \mathbb{R}^K$, where $\ln$ and $\exp$ are element-wise. Hence, both base negentropy yield convex and smooth $L_{\Omega_T}$.

Note that $L_{\Omega_T}$ is not locally strongly convex at every point. We can see this by noting that $T$ is not differentiable and neither is $\Omega_T$, which implies that $\Omega_T^*$ is not strictly convex [80, Theorem 11.13]. This is important for establishing linear surrogate regret bounds (in Section 3.3) because the known square-root regret rate lower bound considers locally strongly convex surrogate losses [39, Theorem 2].

While the loss $L_{\Omega_T}$ is now ensured to be convex and smooth, its gradient calculation, which is the basis for gradient-based optimization [17, 15], remains non-trivial. This is because the gradient of $\min_{\boldsymbol{\pi} \in \Delta^N} \Omega^*(\boldsymbol{\theta} + \mathcal{L}^{\boldsymbol{\rho}}\boldsymbol{\pi})$ contained in (11) cannot be written analytically in general. We show that its gradient calculation reduces to computing $\Pi(\boldsymbol{\theta})$ through the following variant of envelope theorems.

**Lemma 12 (Envelope theorem)** *Suppose Condition 1 holds and $\Omega$ is strictly convex on $\mathrm{dom}(\Omega)$. For any $\boldsymbol{\theta} \in \mathbb{R}^d$ and $\boldsymbol{\pi} \in \Pi(\boldsymbol{\theta})$, we have*

$$\nabla_{\boldsymbol{\theta}}\Big[ \min_{\boldsymbol{\pi} \in \Delta^N} \Omega^*(\boldsymbol{\theta} + \mathcal{L}^{\boldsymbol{\rho}}\boldsymbol{\pi}) \Big] = \nabla\Omega^*(\boldsymbol{\theta} + \mathcal{L}^{\boldsymbol{\rho}}\boldsymbol{\pi}). \tag{13}$$

Deviating from the standard envelope theorems [15, 13], we carefully addresses the non-uniqueness of the minimizers. Given this, the gradient of the convolutional Fenchel–Young loss (11) is accessed via $\nabla_{\boldsymbol{\theta}} L_{\Omega_T}(\boldsymbol{\theta}, y) = \nabla\Omega^*(\boldsymbol{\theta} + \mathcal{L}^{\boldsymbol{\rho}}\boldsymbol{\pi}) - \boldsymbol{\rho}(y)$. At the core of its proof, the differentiability of $\Omega_T^*$ (indirectly obtained via Theorem 11) is vital.

## 3.3 Linear Surrogate Regret Bounds

Now we exhibit linear surrogate regret bounds with the convolutional Fenchel–Young loss.

**Theorem 13 (Linear surrogate regret bound)** *Consider a target loss with $(\boldsymbol{\rho}, \ell^{\boldsymbol{\rho}})$-decomposition. For a negentropy $\Omega : \mathbb{R}^d \to \overline{\mathbb{R}}$, suppose Condition 1 holds. For any $\boldsymbol{\pi}$-argmax link $\varphi$, $(L_{\Omega_T}, \varphi)$ admits the following surrogate regret bound:*

$$\mathtt{Regret}_\ell(\varphi(\boldsymbol{\theta}), \boldsymbol{\eta}) \leq N\mathtt{Regret}_{L_{\Omega_T}}(\boldsymbol{\theta}, \boldsymbol{\eta}), \quad \textit{for any } (\boldsymbol{\theta}, \boldsymbol{\eta}) \in \mathbb{R}^d \times \Delta^K. \tag{14}$$

The regret bound in Theorem 13 indeed has the linear rate $\psi(r) = Nr$ in the form (1), while $L_{\Omega_T}$ can be naturally made convex and smooth as discussed in Section 3.2. This is a remarkable consequence because many authors have previously implied the impossibility to overcome the barrier of the square-root regret rate with convex smooth surrogate losses [50, 76, 39, 8]. The crux of this success lies in the infimal convolution structure in (10), which enables us to additively decompose the conjugate $\Omega_T^*$, exploited below. The constant $N$ will be improved in Section 3.4.

From now on we sketch the regret bound proof. The following lemma is a cornerstone herein.

**Lemma 14 (Lower bound of surrogate regret)** *Assume the same set of conditions as in Theorem 13. For any $\boldsymbol{\theta} \in \mathbb{R}^d$ and $\boldsymbol{\pi} \in \Pi(\boldsymbol{\theta})$, we have the following inequality:*

$$\sum_{t=1}^{N} \pi_t \texttt{Regret}_\ell(t, \boldsymbol{\eta}) \leq \texttt{Regret}_{L_{\Omega_T}}(\boldsymbol{\theta}, \boldsymbol{\eta}), \quad \text{for any } (\boldsymbol{\theta}, \boldsymbol{\eta}) \in \mathbb{R}^d \times \Delta^K. \tag{15}$$

Its proof, formally given in Appendix B, hinges on the following additive regret decomposition:

$$\texttt{Regret}_{L_{\Omega_T}}(\boldsymbol{\theta}, \boldsymbol{\eta}) = \underbrace{R_{L_\Omega}(\boldsymbol{\theta} + \mathcal{L}^{\boldsymbol{\rho}}\boldsymbol{\pi}, \boldsymbol{\eta})}_{\text{Risk of Fenchel–Young loss } L_\Omega \geq 0} + \underbrace{\sum_{t \in \widehat{\mathcal{Y}}} \pi_t \texttt{Regret}_\ell(t, \boldsymbol{\eta})}_{\text{Convex combination of the target regret}} \quad \forall (\boldsymbol{\theta}, \boldsymbol{\pi}) \in \mathbb{R}^d \times \Pi(\boldsymbol{\theta}),$$

which directly implies the lower bound (15) because the Fenchel–Young loss $L_\Omega$ is non-negative. Note that this *additive* decomposition is indispensable for the linear lower bound (15). This becomes possible just because we create the convolutional negentropy $\Omega_T$ based on the additive form in (8), which yields the additive form $\boldsymbol{\theta} + \mathcal{L}^{\boldsymbol{\rho}}\boldsymbol{\pi}$ in the conjugate expression (10). All of these are thanks to the structure of the infimal convolution, and Theorem 13 immediately follows with rescaling.

Finally, we achieve the main result by combining Corollary 11 and Theorem 13. We constructively prove the existence of convex smooth linear-regret surrogate losses, by noting that both strongly convex $\Omega$ satisfying Condition 1 and $\boldsymbol{\pi}$-argmax link do exist.

**Theorem 15 (Main result)** *Consider a target loss $\ell$ with $(\boldsymbol{\rho}, \boldsymbol{\ell}^{\boldsymbol{\rho}})$-decomposition. For a negentropy $\Omega : \mathbb{R}^d \to \overline{\mathbb{R}}$ satisfying Condition 1, suppose that $\Omega$ is strongly convex on $\text{dom}(\Omega)$. Then the convolutional Fenchel–Young loss $L_{\Omega_T}(\cdot, y)$ is convex, differentiable, and smooth over $\mathbb{R}^d$ for any $y \in \mathcal{Y}$. Moreover, with $\boldsymbol{\pi}$-argmax link $\varphi$, $(L_{\Omega_T}, \varphi)$ enjoys the linear surrogate regret bound* (14).

### 3.4 Improving Constant of Linear Surrogate Regret Bounds

The constant $N$ in the linear surrogate regret bound in (14) can be prohibitively large, leading to a vacuous bound. This issue is significant in structured prediction as studied in Osokin et al. [70]. For example, in multilabel classification with the Hamming loss, $N = 2^d$ is the number of all the potential binary label predictions that increases exponentially with $d$, the number of binary labels. In top-$k$ classification, $N = \binom{K}{k}$ is the number of all possible size-$k$ subsets of the class space. Thankfully, the geometry property of the problem (10) provides a promising scheme for reducing the dependency on prediction number $N$, which induces the following improved link.

**Corollary 16 (Improved surrogate regret bound)** *Suppose Condition 1 holds. There exists $\tilde{\boldsymbol{\pi}} : \mathbb{R}^d \to \Delta^N$ such that $\tilde{\boldsymbol{\pi}}(\boldsymbol{\theta}) \in \Pi(\boldsymbol{\theta})$ and $\|\tilde{\boldsymbol{\pi}}(\boldsymbol{\theta})\|_0 \leq \text{affdim}(\mathcal{L}^{\boldsymbol{\rho}}) + 1$ for any $\boldsymbol{\theta} \in \mathbb{R}^d$. Moreover, with the induced $\tilde{\boldsymbol{\pi}}$-argmax link $\varphi_{\tilde{\boldsymbol{\pi}}} : \mathbb{R}^d \to \Delta^N$, $(L_{\Omega_T}, \varphi_{\tilde{\boldsymbol{\pi}}})$ admits the following surrogate regret bound:*

$$\texttt{Regret}_\ell(\varphi_{\tilde{\boldsymbol{\pi}}}(\boldsymbol{\theta}), \boldsymbol{\eta}) \leq [\text{affdim}(\mathcal{L}^{\boldsymbol{\rho}}) + 1] \texttt{Regret}_{L_{\Omega_T}}(\boldsymbol{\theta}, \boldsymbol{\eta}), \quad \text{for any } (\boldsymbol{\theta}, \boldsymbol{\eta}) \in \mathbb{R}^d \times \Delta^K, \tag{16}$$

*where $\text{affdim}(\mathcal{L}^{\boldsymbol{\rho}})$ is the dimension of the affine hull of the column vectors of $\mathcal{L}^{\boldsymbol{\rho}}$.*

Note that we have $\text{affdim}(\mathcal{L}^{\boldsymbol{\rho}}) \leq \min\{N, d\}$, which indicates that the dependency on $N$ can be largely reduced when $d \ll N$. For example, in top-$k$ classification, $\boldsymbol{\rho}(y) = \boldsymbol{e}_y$ is used with $d = K$, which is much smaller than $N = \binom{K}{k}$. In multilabel classification with Hamming loss, $d = \log_2 N$ is logarithmically smaller than $N$ when binary encoding $\boldsymbol{\rho}(y) = \boldsymbol{\nu}(y)$ is used (see Section 2.1).

### 3.5 Bonus: Fisher-consistent Probability Estimator

We discuss a benefit of convex smooth surrogate losses beyond discrete prediction problems. Oftentimes people are interested in a probability estimator over possible prediction outcomes, recovering from surrogate risk minimizers, as in classification with rejection [25, 11, 28, 30, 22, 55, 36]. Herein non-smooth surrogate losses have been unfavored because of lacking reasonable probability estimators [60, 24]. Ramaswamy et al. [76] conjectures the incompatibility of probability estimation with linear regret bounds. By contrast, we give a Fisher-consistent estimator of the linear property $\mathbb{E}_{y \sim \boldsymbol{\eta}}[\boldsymbol{\rho}(y)]$ for convolutional Fenchel–Young losses without sacrificing the linear regret bound.

**Theorem 17 (Fisher-consistent probability estimator)** *Suppose Condition 1 holds and $\Omega$ is strictly convex on $\text{dom}(\Omega)$. For any $\boldsymbol{\eta} \in \text{relint}(\Delta^K)$, the surrogate risk $R_{L_{\Omega_T}}(\cdot, \boldsymbol{\eta})$ is minimized at $\boldsymbol{\theta}^* \in \mathbb{R}^d$ such that $\nabla\Omega^*(\boldsymbol{\theta}^* + \mathcal{L}^{\boldsymbol{\rho}}\boldsymbol{\pi}) = \mathbb{E}_{y \sim \boldsymbol{\eta}}[\boldsymbol{\rho}(y)]$ for any $\boldsymbol{\pi} \in \Pi(\boldsymbol{\theta}^*)$.*

**Algorithm 1** Exact Solution of (17) in $\mathcal{O}(K \ln K)$

---

1: Sort $\boldsymbol{\theta} \in \mathbb{R}^K$ such that $\theta_{(1)} \geq \cdots \geq \theta_{(K)}$.
2: $n \leftarrow \max\{k \in [K] : 1 + k\theta_{(k)} > \sum_{i=1}^k \theta_{(i)}\}$.
3: $\tau(\boldsymbol{\theta}) \leftarrow \frac{\sum_{i=1}^n \theta_{(i)} - 1}{n}$.
4: $\pi_{\log}(\boldsymbol{\theta})_i \leftarrow \max\{\theta_i - \tau(\boldsymbol{\theta}), 0\}$.

---

As a result, the empirical minimizers w.r.t. $L_{\Omega_T}$ are Fisher-consistent estimators of the linear property $\mathbb{E}_{y \sim \boldsymbol{\eta}}[\boldsymbol{\rho}(y)]$. According to the result above, we can first solve $\Pi(\boldsymbol{\theta})$ in (12) and select $\boldsymbol{\pi} \in \Pi(\boldsymbol{\theta})$. Then, $\nabla\Omega^*(\boldsymbol{\theta} + \mathcal{L}^{\rho}\boldsymbol{\pi})$ is a rational estimate of the linear property $\mathbb{E}_{y \sim \boldsymbol{\eta}}[\boldsymbol{\rho}(y)]$. When we follow the trivial loss decomposition (3) with $\boldsymbol{\rho}(y) = \boldsymbol{e}_y$, the estimand is $\mathbb{E}_{y \sim \boldsymbol{\eta}}[\boldsymbol{\rho}(y)] = \boldsymbol{\eta} \in \Delta^K$. For this reason, we say $\nabla\Omega^*(\boldsymbol{\theta}^* + \mathcal{L}^{\rho}\boldsymbol{\pi})$ is a "probability" estimator.

For example, let us recap the encoding of multilabel classification in Section 2.1, where $y \in \mathcal{Y}$ is a multilabel among all $2^d$ possible combinations of $d$ binary labels. A multilabel $y$ is encoded by $\boldsymbol{\nu}(y) \in \{0,1\}^d$, and the Hamming loss is decomposed with $\boldsymbol{\rho}(y) = 1 - 2\boldsymbol{\nu}(y)$. Here $\mathbb{E}_{y \sim \boldsymbol{\eta}}[\boldsymbol{\nu}(y)]$ reads $\bar{\boldsymbol{\nu}} = [\text{Prob}(\boldsymbol{\nu}_i(y) = 1)]_{i=1}^d$, whose $i$-th element indicates the likelihood of the binary class $i$ being positive. Through the relation $\boldsymbol{\rho} = 1 - 2\boldsymbol{\nu}$, the probability estimator $\nabla\Omega^*(\boldsymbol{\theta} + \mathcal{L}^{\rho}\boldsymbol{\pi})$ is capable of recovering $\bar{\boldsymbol{\nu}}$ by $[1 - \nabla\Omega^*(\boldsymbol{\theta} + \mathcal{L}^{\rho}\boldsymbol{\pi})]/2$.

## 4 Example: Multiclass Classification

We demonstrate convolutional Fenchel–Young losses for multiclass classification. and further examples of prediction problems can be found in Appendices D and E, including detailed computational complexity analyses and visualizations of the associated binary classification losses. In multiclass classification, the class space and the prediction space are the same: $\mathcal{Y} = \widehat{\mathcal{Y}} = [K]$. For an input with class probability $\boldsymbol{\eta} \in \Delta^K$, the goal is to predict the most likely class $t \in \text{argmax}_{t \in \mathcal{Y}} \eta_t$. The target loss is the 0-1 loss $\ell_{01}(t, y) = [\![t \neq y]\!]$.

Firstly, we adopt the decomposition (3) for this task, that is, $\boldsymbol{\ell}^{\rho}(t) = [\ell_{01}(t, 1), \cdots, \ell_{01}(t, K)] = \boldsymbol{1} - \boldsymbol{e}_t$ and $\boldsymbol{\rho}(y) = \boldsymbol{e}_y$, which corresponds to the one-hot encoding commonly used in this task. In this case, $N = d = K$, and $\mathcal{L}^{\rho} = \boldsymbol{1}\boldsymbol{1}^\top - I$.

Next, we move on to the convolutional negentropy $\Omega_T$. For the choice of $\Omega$, we use the Shannon negentropy, which induces the celebrated cross-entropy loss in the original Fenchel–Young loss framework [16, Table 1]. Its conjugate is the log-sum-exp function $\Omega^*(\boldsymbol{\theta}) = \ln\langle\exp(\boldsymbol{\theta}), \boldsymbol{1}\rangle$. Then we can calculate the conjugate of $\Omega_T$ based on Lemma 8:

$$\Omega_T^*(\boldsymbol{\theta}) = \min_{\boldsymbol{\pi} \in \Delta^K} \Omega^*(\boldsymbol{\theta} + \mathcal{L}^{\rho}\boldsymbol{\pi}) = \min_{\boldsymbol{\pi} \in \Delta^K} \ln\langle\exp(\boldsymbol{\theta} + \boldsymbol{1} - \boldsymbol{\pi}), \boldsymbol{1}\rangle. \tag{17}$$

Since $\Omega$ is proper convex and l.s.c. with $\text{conv}(\boldsymbol{\rho}(\mathcal{Y})) = \Delta^K = \text{dom}(\Omega)$ and $\text{dom}(\Omega^*) = \mathbb{R}^K$, $\Omega$ satisfies Condition 1. Furthermore, since $\Omega$ is strongly convex on $\Delta^K$, we can finally derive the convolutional Fenchel–Young loss (11) by nothing $\Omega_T(\boldsymbol{\rho}(y)) = 0$ for all $y \in \mathcal{Y}$, as follows:

$$L_{\Omega_T}(\boldsymbol{\theta}, y) = \Omega_T^*(\boldsymbol{\theta}) + \Omega_T(\boldsymbol{\rho}(y)) - \langle\boldsymbol{\theta}, \boldsymbol{\rho}(y)\rangle = \min_{\boldsymbol{\pi} \in \Delta^K} \ln\langle\exp(\boldsymbol{\theta} + \boldsymbol{1} - \boldsymbol{\pi}), \boldsymbol{1}\rangle - \theta_y, \tag{18}$$

which is convex and smooth in $\boldsymbol{\theta}$ thanks to Theorem 11. While shares the form of cross-entropy loss $L_{\Omega}(\boldsymbol{\theta}, y) = \ln\langle\exp(\boldsymbol{\theta}), \boldsymbol{1}\rangle - \theta_y$, it further incorporates an additional bounded perturbation.

Solving the minimization problem (17) is important from the computational aspects, including the gradient calculation of $L_{\Omega_T}$ and accessing to the probability estimator given by Theorem 17. We provide Algorithm 1 to solve (17) with $\mathcal{O}(K \ln K)$ time.

**Lemma 18** *For any $\boldsymbol{\theta} \in \mathbb{R}^K$, the problem* (17) *has a unique minimizer $\boldsymbol{\pi}_{\log}(\boldsymbol{\theta}) \in \Pi(\boldsymbol{\theta})$, which can be obtained in $\mathcal{O}(K \ln K)$ time by Algorithm 1.*

The proof is deferred to Appendix C.1, following from the KKT conditions. Eventually we have

$$\nabla\Omega_T^*(\boldsymbol{\theta}) = \text{softmax}\left(\boldsymbol{\theta} + \boldsymbol{1} - \boldsymbol{\pi}_{\log}(\boldsymbol{\theta})\right), \quad \text{where } \text{softmax}(\boldsymbol{\theta})_y = \frac{\exp(\theta_y)}{\sum_{i=1}^K \exp(\theta_i)} \quad \text{for } y \in [K].$$

Algorithm 1 comes with a significant resemblance with the sparsemax [58, Algorithm 1], which is determined by the similar structure shared by (17) and Euclidean projection problem [58, (2)]. Since $\Pi(\boldsymbol{\theta})$ is a singleton, we have the unique $\boldsymbol{\pi}_{\log}$-argmax link $\varphi_{\boldsymbol{\pi}_{\log}}$ (Definition 10). While we need to solve problem (17) for every $\boldsymbol{\theta}$ to have access to $\nabla_{\boldsymbol{\theta}} L_{\Omega_T}(\boldsymbol{\theta}, y)$, the prediction by the $\boldsymbol{\pi}_{\log}$-argmax link is much cheaper, without requiring Algorithm 1, which indicates that we can simply use the class label with the largest score $\max_{y \in [K]} \theta_y$ in test time. The proof can be found in Appendix C.2.

**Proposition 19** *A prediction link $\varphi$ is the $\boldsymbol{\pi}_{\log}$-argmax link if and only if $\varphi(\boldsymbol{\theta}) \in \operatorname{argmax}_{t \in \mathcal{Y}} \theta_t$.*

**Remark 20** *Under classification, a surrogate loss with a regret bound is classification-calibrated, which indicates that the surrogate risk minimization eventually leads to the Bayes-optimal classifier. In literature, Blondel [14] and Wang and Scott [94] have investigated sufficient conditions for Fenchel–Young losses to be classification-calibrated. Therein the base negentropy is assumed to be of Legendre-type or twice differentiable. Our convolutional Fenchel–Young losses are interesting because we do not require these conditions to yield both the smoothness and a regret bound. Thus it remains open to relax these existing sufficient conditions for Fenchel–Young losses further.*

## 5 Discussion

**Convex non-smooth linear regret losses.** Compared to existing convex non-smooth surrogates with linear regret, e.g., polyhedral losses [37], our smooth convex surrogate has several advantages. First, smoothness enables more efficient optimization, as supported by results in both deterministic and stochastic optimization regimes [67, 83] while it is left open to exploit specific structures arising from polyhedral losses to achieve faster optimization. In addition, the smoothness can lead to optimistic ERM rates of estimation error [87], offering better estimation in easier tasks. Another appealing aspect is that our loss admits a consistent probability estimator (Theorem 17), which can be valuable for downstream tasks such as uncertainty quantification and calibration.

**Efficient gradient calculation.** In general discrete prediction problems, we need to solve the minimization (10) to take a gradient of $L_{\Omega_T}$, which is potentially demanding over a high-dimensional domain $\Delta^N$. To have access to the gradient, we can alternatively solve

$$\min_{\boldsymbol{\nu} \in V} \Omega^*(\boldsymbol{\theta} + \boldsymbol{\nu}), \quad \text{where} \quad V := \operatorname{conv}\Big(\{\boldsymbol{\ell}^{\boldsymbol{\rho}}(t)\}_{t \in [N]}\Big). \tag{19}$$

To see this, let us denote the minimizer of (19) by $\boldsymbol{\nu}^*$. Then (19) is an equivalent optimization problem to (10) because there exists $\boldsymbol{\pi} \in \Pi(\boldsymbol{\theta})$ such that $\boldsymbol{\nu}^* = \mathcal{L}^{\boldsymbol{\rho}} \boldsymbol{\pi}$. Eventually $\nabla \Omega^*(\boldsymbol{\theta} + \boldsymbol{\nu}^*)$ serves as an alternative to the gradient formula (13). Thus we can reduce the dimensionality of the optimization problem. For example, in multilabel classification (Section 2.1), $V = [0, 1]^d$ has a logarithmically smaller optimization dimensionality than $N = 2^d$. We can solve (19) with this box constraint efficiently by using the standard L-BFGS solver [47].

**Randomized prediction link.** Recall that the $\boldsymbol{\pi}$-argmax link $\varphi$ (Definition 10) deterministically outputs in the probability simplex $\Delta^N$. Instead we can define a randomized link $\tilde{\varphi}$ such that $\Pr[\tilde{\varphi}(\boldsymbol{\theta}) = t] = \pi_t(\boldsymbol{\theta})$. This yields a better regret bound by Lemma 14, as follows:

$$\mathbb{E}_{\tilde{\varphi}(\boldsymbol{\theta})}[\mathtt{Regret}_\ell(\tilde{\varphi}(\boldsymbol{\theta}), \boldsymbol{\eta})] = \sum_{t=1}^N \pi_t(\boldsymbol{\theta}) \mathtt{Regret}_\ell(t, \boldsymbol{\eta}) \le \mathtt{Regret}_{L_{\Omega_T}}(\boldsymbol{\theta}, \boldsymbol{\eta}) \quad \forall(\boldsymbol{\theta}, \boldsymbol{\eta}) \in \mathbb{R}^d \times \Delta^K,$$

where the expectation is taken over the randomness of $\tilde{\varphi}$. Although we adopt the *expected* target regret differently from Theorem 13, the constant of regret bounds is strikingly improved from $N$ to 1, which is dimension-free. Sakaue et al. [82] observes a similar regret improvement by a randomized link for online structured prediction with Fenchel–Young losses.

## 6 Conclusion

In this work, we construct convex and smooth surrogate losses with linear surrogate regret bounds by leveraging Fenchel–Young losses and infimal convolution. Our results demonstrate that convexity, smoothness, and linear surrogate regret are compatible for arbitrary discrete prediction problems. Moreover, our loss naturally admits consistent probability estimator, bridging the gap between linear regret and estimation. We illustrate the broad applicability of our approach through examples in multiclass classification, classification with rejection, and multilabel ranking. Overall, this study highlights the utility of convex analysis as a principled tool for designing surrogate losses.

## Acknowledgement

We thank Shinsaku Sakaue for his valuable insights on the target-loss decomposition and for his careful review, and we also thank the anonymous reviewers for their attentive reading of the manuscript and their many thoughtful comments and suggestions. This research is supported by the Ministry of Education, Singapore, under its MOE AcRF Tier 2 Award MOE-T2EP20223-0003. Any opinions, findings and conclusions or recommendations expressed in this material are those of the author(s) and do not reflect the views of the Ministry of Education, Singapore. YC is also supported by Google PhD Fellowship program. HB is supported by JST PRESTO (Grant No. JPMJPR24K6), Japan. Lei Feng is supported by the Big Data Computing Center of Southeast University.

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

# A    Additional Discussions on Related Losses

**Fast rate losses.**    Let us briefly discuss the target risk estimation error rates induced by different surrogate losses. Suppose that a surrogate risk can be estimated with the estimation error upper bound $\epsilon_{\text{est}}$ and the surrogate regret rate is $\psi$. Then, the target risk estimation error is of order $\psi(\epsilon_{\text{est}})$. Hence, we need to take care of both $\psi$ and $\epsilon_{\text{est}}$ to discuss the target risk estimation.

On the one hand, loss functions entailing either strongly convex or exponentially concave are known to achieve fast estimation error rates with ERM (e.g., $\mathcal{O}(1/n)$) in standard parametric setting, where the hypothesis class is of finite dimension [87, 91, 61, 92]. However, Frongillo and Waggoner [39, Theorem 4] reveals that typical fast rate losses suffer at least from square-root regret rates, and thus their corresponding target risk estimation error bounds are $\mathcal{O}(1/\sqrt{n})$ at least. On the other hand, convolutional Fenchel–Young losses (which is convex smooth but not strongly convex) yield the estimation error upper bounds of order $\mathcal{O}(1/\sqrt{n})$ under the same setting, while the final target risk upper bounds are in order of $\mathcal{O}(1/\sqrt{n})$ thanks to the linear regret rate function. As a result, convolutional Fenchel–Young losses achieve target regret convergence rates that are comparable to those of existing fast-rate losses.

In more general nonparametric settings, where fast rates are often hard to achieve without additional assumptions [62], convolutional Fenchel–Young losses achieve a target risk estimation error bound of order $\mathcal{O}(1/\sqrt{n})$. In contrast, strongly convex surrogate losses typically achieve a slower target risk estimation error rate than $\mathcal{O}(1/\sqrt{n})$ because the general nonparametric estimation error is $\mathcal{O}(1/\sqrt{n})$ but the regret rate are slower than $\psi(r) = \mathcal{O}(r)$.

While we do not intend to argue that the convolutional Fenchel–Young loss is always better, we would like to highlight that it may be a good alternative when the parametric conditions for fast rates cannot be readily justified. It also remains an open and worthwhile question whether fast rates can be obtained for our loss under additional assumptions, such as low-noise or margin conditions.

**Smooth non-convex linear regret losses.**    While our focus is on convex surrogates due to their favorable optimization and statistical properties, we note that certain smooth but non-convex surrogates, such as the mean absolute error (MAE) [41, 52] and structured comp-sum losses with MAE [53], also achieve linear regret. These methods, while typically non-convex and not equipped with probability estimator, offer valuable advantages in other aspects, such as robustness to label noise, $\mathcal{H}$-consistency, and potential benefits under adversarial conditions [10, 6], where non-convexity can play a meaningful role.

# B    Deferred Proofs in Section 3

Recalling the definition of $T(\boldsymbol{p}) = -\min_{t \in \widehat{\mathcal{Y}}} \langle \boldsymbol{p}, \boldsymbol{\ell}^{\boldsymbol{\rho}}(t) \rangle$, which is the negative of the affinely transformed target Bayes risk in (9). It can be inferred that it is proper convex and l.s.c., and we have $\Omega_T = \Omega + T$.

## B.1    Proof of Lemma 8

**Proof.** Since $\Omega$ and $T$ are proper convex and l.s.c., so are $\Omega_T$ and thus $\Omega_T^*$. Then we prove (10) using the infimal convolution. First, by noting that $T$ is nothing else but the support function of the closed convex set $\text{conv}(\{-\boldsymbol{\ell}^{\boldsymbol{\rho}}(t)\}_{t \in \widehat{\mathcal{Y}}})$, we can express $T^*$ as follows:

$$T^*(\boldsymbol{\theta}) = \sup_{\boldsymbol{p} \in \mathbb{R}^d} \left[ \langle \boldsymbol{\theta}, \boldsymbol{p} \rangle - \max_{t \in \widehat{\mathcal{Y}}} \langle \boldsymbol{p}, -\boldsymbol{\ell}^{\boldsymbol{\rho}}(t) \rangle \right] = \mathbb{I}_{\text{conv}\left(\{-\boldsymbol{\ell}^{\boldsymbol{\rho}}(t)\}_{t \in \widehat{\mathcal{Y}}}\right)}(\boldsymbol{\theta}),$$

where we use the conjugacy relationship between a support function and indicator function of a closed convex set $\text{conv}(\{-\boldsymbol{\ell}^{\boldsymbol{\rho}}(t)\}_{t \in \widehat{\mathcal{Y}}})$ [79, Section 13]. According to Condition 1 and the definition of $T$, we see that both $\Omega$ and $T$ are proper convex and $T$ is continuous on $\text{dom}(\Omega)$. Then we use the infimal convolution [20] to derive the conjugate of $\Omega_T$:

$$\Omega_T^*(\boldsymbol{\theta}) = \inf_{\boldsymbol{\theta}' \in \mathbb{R}^d} \left[ \Omega^*(\boldsymbol{\theta} - \boldsymbol{\theta}') + \mathbb{I}_{\text{conv}\left(\{-\boldsymbol{\ell}^{\boldsymbol{\rho}}(t)\}_{t \in \widehat{\mathcal{Y}}}\right)}(\boldsymbol{\theta}') \right]$$

$$= \inf_{\boldsymbol{\theta}' \in \text{conv}\left(\{-\boldsymbol{\ell}^{\boldsymbol{\rho}}(t)\}_{t \in \widehat{\mathcal{Y}}}\right)} \Omega^*(\boldsymbol{\theta} - \boldsymbol{\theta}')$$

$$= \inf_{\boldsymbol{\pi} \in \Delta^N} \Omega^*(\boldsymbol{\theta} + \mathcal{L}^{\boldsymbol{\rho}}\boldsymbol{\pi}).$$

Next we show that the infimum is indeed achieved by some $\boldsymbol{\pi} \in \Delta^N$. Note that for any $\boldsymbol{\theta} \in \mathbb{R}^d$, the set $\Theta := \{\boldsymbol{\theta} + \mathcal{L}^{\boldsymbol{\rho}}\boldsymbol{\pi} : \boldsymbol{\pi} \in \Delta^N\}$ is compact and non-empty, and that $\Omega^*$ is l.s.c. on $\Theta$ since $\mathrm{dom}(\Omega^*) = \mathbb{R}^d$. Then the infimum of $\Omega^*(\boldsymbol{\theta})$ is achieved by some $\boldsymbol{\theta}'' \in \Theta$, and there must exist $\boldsymbol{\pi} \in \Delta^N$ such that $\boldsymbol{\theta} + \mathcal{L}^{\boldsymbol{\rho}}\boldsymbol{\pi} = \boldsymbol{\theta}''$ by the definition of $\Theta$, which complete the proof of the existence of the minimizer.

Finally we see that for any $\boldsymbol{\theta} \in \mathbb{R}^d$, there exists $\boldsymbol{\pi} \in \Delta^N$ such that $\Omega_T^*(\boldsymbol{\theta}) = \Omega^*(\boldsymbol{\theta} + \mathcal{L}^{\boldsymbol{\rho}}\boldsymbol{\pi})$, which is smaller than $+\infty$ since $\mathrm{dom}(\Omega^*) = \mathbb{R}^d$ and $\boldsymbol{\theta} + \mathcal{L}^{\boldsymbol{\rho}}\boldsymbol{\pi} \in \mathbb{R}^d$. This indicates that $\mathrm{dom}(\Omega_T^*) = \mathbb{R}^d$ because $\boldsymbol{\theta}$ is chosen arbitrarily. ∎

## B.2 Proof of Theorem 11

**Proof.** In the convolutional Fenchel–Young loss (11), the term $\Omega_T(\boldsymbol{\rho}(y)) - \langle \boldsymbol{\theta}, \boldsymbol{\rho}(y) \rangle$ is linear. Hence it suffices to prove the convexity and smoothness of the conjugate $\min_{\boldsymbol{\pi} \in \Delta^N} \Omega^*(\boldsymbol{\theta} + \mathcal{L}^{\boldsymbol{\rho}}\boldsymbol{\pi})$.

When $\Omega$ is strictly convex, $\Omega_T$ is also strictly convex since $T$ is convex. Since the strict convexity holds on $\mathrm{dom}(\Omega_T)$, $\Omega_T$ is further essentially strictly convex, that is, strictly convex on every convex subset of $\mathrm{dom}(\partial\Omega_T)$ [79, p253]. According to Rockafellar [79, Theorem 26.3], $\Omega_T^*$ is essentially smooth, that is, differentiable throughout non-empty $\mathrm{int}(\mathrm{dom}(\Omega_T^*))$ with $\|\nabla\Omega_T^*(\boldsymbol{\theta})\|$ diverging to $+\infty$ when $\boldsymbol{\theta}$ approaches a boundary point of $\mathrm{int}(\mathrm{dom}(\Omega_T^*))$, which immediately indicates the differentiability on $\mathbb{R}^d$.

When $\Omega$ is further strongly convex, so is $\Omega_T$. According to Condition 1 and Lemma 8, $\Omega_T$ is proper convex and l.s.c., and thus the biconjugate $\Omega_T^{**}$ matches $\Omega_T$ by the Fenchel–Moreau theorem. According to Rockafellar and Wets [80, Proposition 12.60], $\Omega_T^*$ is smooth since its conjugate $\Omega_T^{**} = \Omega_T$ is strongly convex. ∎

## B.3 Proof of Lemma 12

**Proof.** According to Condition 1, the strict convexity of $\Omega$, and the proof of Theorem 11, both $\Omega_T^*$ and $\Omega^*$ are differentiable on $\mathbb{R}^d$. In addition, we have $\mathrm{dom}(\Omega) = \mathrm{dom}(\Omega_T)$ according to (8).

Note that $\Omega^*(\boldsymbol{\theta} + \mathcal{L}^{\boldsymbol{\rho}}\boldsymbol{\pi})$ is convex in $\boldsymbol{\pi}$. Then its first-order optimal condition for any $\boldsymbol{\pi} \in \Pi(\boldsymbol{\theta})$ reads

$$\langle \nabla\Omega^*(\boldsymbol{\theta} + \mathcal{L}^{\boldsymbol{\rho}}\boldsymbol{\pi}), \mathcal{L}^{\boldsymbol{\rho}}(\boldsymbol{\pi}' - \boldsymbol{\pi}) \rangle \geq 0 \qquad \text{any } \boldsymbol{\pi}' \in \Delta^N.$$

First, for any $\boldsymbol{\pi} \in \Pi(\boldsymbol{\theta})$, we choose

$$t \in \mathrm{argmin}_{t' \in \widehat{\mathcal{Y}}} \langle \nabla\Omega^*(\boldsymbol{\theta} + \mathcal{L}^{\boldsymbol{\rho}}\boldsymbol{\pi}), \boldsymbol{\ell}^{\boldsymbol{\rho}}(t') \rangle,$$

then we have

$$\begin{aligned}
\Omega^*(\boldsymbol{\theta} + \mathcal{L}^{\boldsymbol{\rho}}\boldsymbol{\pi}) &\overset{(A)}{=} (\boldsymbol{\theta} + \mathcal{L}^{\boldsymbol{\rho}}\boldsymbol{\pi})^\top \nabla\Omega^*(\boldsymbol{\theta} + \mathcal{L}^{\boldsymbol{\rho}}\boldsymbol{\pi}) - \Omega(\nabla\Omega^*(\boldsymbol{\theta} + \mathcal{L}^{\boldsymbol{\rho}}\boldsymbol{\pi})) \\
&= \boldsymbol{\theta}^\top \nabla\Omega^*(\boldsymbol{\theta} + \mathcal{L}^{\boldsymbol{\rho}}\boldsymbol{\pi}) + \langle \nabla\Omega^*(\boldsymbol{\theta} + \mathcal{L}^{\boldsymbol{\rho}}\boldsymbol{\pi}), \mathcal{L}^{\boldsymbol{\rho}}\boldsymbol{\pi} \rangle - \Omega(\nabla\Omega^*(\boldsymbol{\theta} + \mathcal{L}^{\boldsymbol{\rho}}\boldsymbol{\pi})) \\
&\overset{(A)}{\leq} \boldsymbol{\theta}^\top \nabla\Omega^*(\boldsymbol{\theta} + \mathcal{L}^{\boldsymbol{\rho}}\boldsymbol{\pi}) + \langle \nabla\Omega^*(\boldsymbol{\theta} + \mathcal{L}^{\boldsymbol{\rho}}\boldsymbol{\pi}), \mathcal{L}^{\boldsymbol{\rho}}\boldsymbol{e}_t \rangle - \Omega(\nabla\Omega^*(\boldsymbol{\theta} + \mathcal{L}^{\boldsymbol{\rho}}\boldsymbol{\pi})) \\
&= \boldsymbol{\theta}^\top \nabla\Omega^*(\boldsymbol{\theta} + \mathcal{L}^{\boldsymbol{\rho}}\boldsymbol{\pi}) + \langle \nabla\Omega^*(\boldsymbol{\theta} + \mathcal{L}^{\boldsymbol{\rho}}\boldsymbol{\pi}), \boldsymbol{\ell}^{\boldsymbol{\rho}}(t) \rangle - \Omega(\nabla\Omega^*(\boldsymbol{\theta} + \mathcal{L}^{\boldsymbol{\rho}}\boldsymbol{\pi})) \\
&\overset{(B)}{=} \boldsymbol{\theta}^\top \nabla\Omega^*(\boldsymbol{\theta} + \mathcal{L}^{\boldsymbol{\rho}}\boldsymbol{\pi}) + \underbrace{\min_{t' \in \widehat{\mathcal{Y}}} \langle \nabla\Omega^*(\boldsymbol{\theta} + \mathcal{L}^{\boldsymbol{\rho}}\boldsymbol{\pi}), \boldsymbol{\ell}^{\boldsymbol{\rho}}(t') \rangle}_{= -T(\nabla\Omega^*(\boldsymbol{\theta} + \mathcal{L}^{\boldsymbol{\rho}}\boldsymbol{\pi}))} - \Omega(\nabla\Omega^*(\boldsymbol{\theta} + \mathcal{L}^{\boldsymbol{\rho}}\boldsymbol{\pi})) \\
&\overset{(C)}{=} \boldsymbol{\theta}^\top \nabla\Omega^*(\boldsymbol{\theta} + \mathcal{L}^{\boldsymbol{\rho}}\boldsymbol{\pi}) - \Omega_T(\nabla\Omega^*(\boldsymbol{\theta} + \mathcal{L}^{\boldsymbol{\rho}}\boldsymbol{\pi})) \\
&\overset{(D)}{\leq} \sup_{\boldsymbol{p} \in \mathrm{dom}(\Omega_T)} \boldsymbol{\theta}^\top \boldsymbol{p} - \Omega_T(\boldsymbol{p}) \\
&= \Omega_T^*(\boldsymbol{\theta}),
\end{aligned}$$

where (A) owes to the optimality of $\boldsymbol{\pi} \in \Pi(\boldsymbol{\theta})$, (B) holds by the definition of $t$, and (C) holds by the definition of the convolutional negentropy $\Omega_T$ (8). Since $\boldsymbol{\pi} \in \Pi(\boldsymbol{\theta})$, we have $\Omega^*(\boldsymbol{\theta} + \mathcal{L}^{\boldsymbol{\rho}}\boldsymbol{\pi}) = \Omega_T^*(\boldsymbol{\theta})$ according to Lemma 8. Thus the above inequality is indeed an identity. In particular, (D) becomes an identity, which implies that the supremum of $\sup_{\boldsymbol{p} \in \mathrm{dom}(\Omega_T)} \boldsymbol{\theta}^\top \boldsymbol{p} - \Omega_T(\boldsymbol{p})$ is achieved at $\nabla\Omega^*(\boldsymbol{\theta} + \mathcal{L}^{\boldsymbol{\rho}}\boldsymbol{\pi})$. Since the supremum is also achieved at $\nabla\Omega_T^*(\boldsymbol{\theta})$ and the maximizer is unique according to Rockafellar and Wets [80, Proposition 11.3] and the differentiability of $\Omega_T^*$, we have

$$\nabla\Omega_T^*(\boldsymbol{\theta}) = \nabla\Omega^*(\boldsymbol{\theta} + \mathcal{L}^{\boldsymbol{\rho}}\boldsymbol{\pi}).$$

Since $\boldsymbol{\pi} \in \Pi(\boldsymbol{\theta})$ and $\boldsymbol{\theta} \in \mathbb{R}^d$ are chosen arbitrarily, this concludes the proof. ∎

## B.4 Proof of Theorem 13

**Proof.** Fix any $\boldsymbol{\theta} \in \mathbb{R}^d$ and choose any $\boldsymbol{\pi} \in \Delta^N$ out of $\Pi(\boldsymbol{\theta})$. Then we have

$$\forall \boldsymbol{\eta} \in \Delta^K, \ \ \mathrm{Regret}_{L_{\Omega_T}}(\boldsymbol{\theta}, \boldsymbol{\eta}) \geq \sum_{t=1}^N \pi_t \mathrm{Regret}_\ell(t, \boldsymbol{\eta}) \quad \text{(by (15))}$$
$$\geq \pi_{\varphi(\boldsymbol{\theta})} \mathrm{Regret}_\ell(\varphi(\boldsymbol{\theta}), \boldsymbol{\eta}) \quad \text{(because } \pi_t \geq 0 \text{ for any } t \in \widehat{\mathcal{Y}})$$
$$\geq \mathrm{Regret}_\ell(\varphi(\boldsymbol{\theta}), \boldsymbol{\eta})/N. \quad \text{(by definition of } \varphi \text{ in Defintion 10)}$$

∎

## B.5 Proof of Lemma 14

**Proof.** First, we derive the Bayes risk of the convolutional Fenchel–Young loss $L_{\Omega_T}$ as follows:

$$\underline{R}_{L_{\Omega_T}}(\boldsymbol{\eta}) = \inf_{\boldsymbol{\theta} \in \mathbb{R}^d} R_{L_{\Omega_T}}(\boldsymbol{\theta}, \boldsymbol{\eta})$$
$$= \inf_{\boldsymbol{\theta} \in \mathbb{R}^d} \Big[ \Omega_T^*(\boldsymbol{\theta}) - \langle \boldsymbol{\theta}, \mathbb{E}_{y \sim \boldsymbol{\eta}}[\boldsymbol{\rho}(y)] \rangle \Big] + \mathbb{E}_{y \sim \boldsymbol{\eta}}[\Omega_T(\boldsymbol{\rho}(y))]$$
$$= - \sup_{\boldsymbol{\theta} \in \mathbb{R}^d} \Big[ \langle \boldsymbol{\theta}, \mathbb{E}_{y \sim \boldsymbol{\eta}}[\boldsymbol{\rho}(y)] \rangle - \Omega_T^*(\boldsymbol{\theta}) \Big] + \mathbb{E}_{y \sim \boldsymbol{\eta}}[\Omega_T(\boldsymbol{\rho}(y))]$$
$$= -\Omega_T^{**}\Big( \mathbb{E}_{y \sim \boldsymbol{\eta}}[\boldsymbol{\rho}(y)] \Big) + \mathbb{E}_{y \sim \boldsymbol{\eta}}[\Omega_T(\boldsymbol{\rho}(y))]$$
$$= \mathbb{E}_{y \sim \boldsymbol{\eta}}[\Omega_T(\boldsymbol{\rho}(y))] - \Omega_T\Big( \mathbb{E}_{y \sim \boldsymbol{\eta}}[\boldsymbol{\rho}(y)] \Big)$$

where the Fenchel–Moreau theorem is applied to proper convex and l.s.c. $\Omega_T^*$ (by Lemma 8) at the last identity. Then we can get the following regret lower bound for any $\boldsymbol{\pi} \in \Pi(\boldsymbol{\theta})$:

$$\mathrm{Regret}_{L_{\Omega_T}}(\boldsymbol{\theta}, \boldsymbol{\eta}) = R_{L_{\Omega_T}}(\boldsymbol{\theta}, \boldsymbol{\eta}) - \underline{R}_{L_{\Omega_T}}(\boldsymbol{\eta})$$
$$= \Omega_T^*(\boldsymbol{\theta}) - \Big\langle \boldsymbol{\theta}, \mathbb{E}_{y \sim \boldsymbol{\eta}}[\boldsymbol{\rho}(y)] \Big\rangle + \Omega_T\Big( \mathbb{E}_{y \sim \boldsymbol{\eta}}[\boldsymbol{\rho}(y)] \Big)$$
$$= \Omega^*(\boldsymbol{\theta} + \mathcal{L}^{\boldsymbol{\rho}}\boldsymbol{\pi}) - \Big\langle \boldsymbol{\theta}, \mathbb{E}_{y \sim \boldsymbol{\eta}}[\boldsymbol{\rho}(y)] \Big\rangle + \Omega\Big( \mathbb{E}_{y \sim \boldsymbol{\eta}}[\boldsymbol{\rho}(y)] \Big) + T\Big( \mathbb{E}_{y \sim \boldsymbol{\eta}}[\boldsymbol{\rho}(y)] \Big)$$
$$\stackrel{\text{(A)}}{=} \underbrace{\Omega^*(\boldsymbol{\theta} + \mathcal{L}^{\boldsymbol{\rho}}\boldsymbol{\pi}) - \Big\langle \boldsymbol{\theta} + \mathcal{L}^{\boldsymbol{\rho}}\boldsymbol{\pi}, \mathbb{E}_{y \sim \boldsymbol{\eta}}[\boldsymbol{\rho}(y)] \Big\rangle + \Omega\Big( \mathbb{E}_{y \sim \boldsymbol{\eta}}[\boldsymbol{\rho}(y)] \Big)}_{\geq 0 \quad \text{due to the Fenchel–Young inequality}}$$
$$+ \Big\langle \mathcal{L}^{\boldsymbol{\rho}}\boldsymbol{\pi}, \mathbb{E}_{y \sim \boldsymbol{\eta}}[\boldsymbol{\rho}(y)] \Big\rangle + T\Big( \mathbb{E}_{y \sim \boldsymbol{\eta}}[\boldsymbol{\rho}(y)] \Big)$$
$$\stackrel{\text{(B)}}{\geq} \sum_{t=1}^N \pi_t \Big\langle \mathbb{E}_{y \sim \boldsymbol{\eta}}[\boldsymbol{\rho}(y)], \boldsymbol{\ell}^{\boldsymbol{\rho}}(t) \Big\rangle - \min_{t \in \widehat{\mathcal{Y}}} \Big\langle \mathbb{E}_{y \sim \boldsymbol{\eta}}[\boldsymbol{\rho}(y)], \boldsymbol{\ell}^{\boldsymbol{\rho}}(t) \Big\rangle$$
$$\stackrel{\text{(C)}}{=} \sum_{t=1}^N \pi_t \mathrm{Regret}_\ell(t, \boldsymbol{\eta}),$$

where (A) holds according to the explicit form of $\Omega_T^*$ in (10) and the definition of $\Pi(\boldsymbol{\theta})$, (B) owes to Fenchel–Young inequality, and (C) holds according to the definition of $(\boldsymbol{\rho}, \boldsymbol{\ell}^{\boldsymbol{\rho}})$-decomposition in (4). This concludes the proof. ∎

### B.6 Proof of Corollary 16

**Proof.** Fix an arbitrary $\boldsymbol{\theta} \in \mathbb{R}^d$. Since $\Pi(\boldsymbol{\theta})$ is non-empty, we can select $\boldsymbol{\pi}$ from $\Pi(\boldsymbol{\theta})$. Note that $\mathcal{L}^{\boldsymbol{\rho}}\boldsymbol{\pi} \in \text{conv}(\{\boldsymbol{\ell}^{\boldsymbol{\rho}}(t)\}_{t \in \widehat{\mathcal{Y}}})$. According to Carathéodory's theorem, there exists $\tilde{\boldsymbol{\pi}}$ such that $\|\tilde{\boldsymbol{\pi}}\|_0 \leq$ affdim$(\mathcal{L}^{\boldsymbol{\rho}}) + 1$ and $\mathcal{L}^{\boldsymbol{\rho}}\tilde{\boldsymbol{\pi}} = \mathcal{L}^{\boldsymbol{\rho}}\boldsymbol{\pi}$ is also in $\Pi(\boldsymbol{\theta})$. Since $\boldsymbol{\theta}$ is chosen arbitrarily, we can then construct a single-valued function $\tilde{\boldsymbol{\pi}} : \mathbb{R}^d \to \Delta^N$ such that $\tilde{\boldsymbol{\pi}}(\boldsymbol{\theta}) \in \Pi(\boldsymbol{\theta})$ and $\|\tilde{\boldsymbol{\pi}}(\boldsymbol{\theta})\|_0 \leq$ affdim$(\mathcal{L}^{\boldsymbol{\rho}}) + 1$. Noting that $\max_{t \in \widehat{\mathcal{Y}}} \tilde{\pi}_t(\boldsymbol{\theta}) > 1/[\text{affdim}(\mathcal{L}^{\boldsymbol{\rho}}) + 1]$, we can complete the rest of the proof similarly to Theorem 13. ∎

**Remark 21** *Interestingly, Ramaswamy and Agarwal [73] also used the affine dimension of the loss matrix to study the dimension of convex surrogates, suggesting that this concept plays a important role in loss function design and deserves more attention.*

### B.7 Proof of Theorem 17

**Proof.** First of all, we prove that $\text{relint}(\text{conv}(\boldsymbol{\rho}(\mathcal{Y})))$ is in the image of $\nabla\Omega_T^*$ by contradiction. According to Rockafellar [79, Theorem 23.4], we have that $\partial\Omega_T(\boldsymbol{p})$ is non-empty for all $\boldsymbol{p} \in$ relint$(\text{dom}(\Omega_T))$. Now suppose there exists $\boldsymbol{p} \in$ relint$(\text{dom}(\Omega_T))$ such that $\boldsymbol{p} \neq \nabla\Omega_T^*(\boldsymbol{\theta})$ for any $\boldsymbol{\theta} \in \mathbb{R}^d$. Since $\partial\Omega_T(\boldsymbol{p})$ is non-empty, there exists $\boldsymbol{\theta}' \in \mathbb{R}^d$ such that $\boldsymbol{\theta}' = \nabla\Omega_T(\boldsymbol{p})$. By Rockafellar and Wets [80, Proposition 11.3] (applied on proper l.s.c. $\Omega_T$), we have $\boldsymbol{p} = \nabla\Omega_T^*(\boldsymbol{\theta}')$, which contradicts the assumption $\boldsymbol{p} \neq \nabla\Omega_T^*(\boldsymbol{\theta}')$. Thus we have verified that $\text{relint}(\text{dom}(\Omega_T))$ is in the image of $\nabla\Omega_T^*$. This additionally implies that $\text{relint}(\text{conv}(\boldsymbol{\rho}(\mathcal{Y}))) \subseteq \text{relint}(\text{dom}(\Omega_T))$ is also in the image of $\nabla\Omega_T^*$ because of Condition 1.

Now we fix $\boldsymbol{\eta} \in \text{relint}(\Delta^K)$ and note that

$$R_{L_{\Omega_T}}(\boldsymbol{\theta}, \boldsymbol{\eta}) = \Omega_T^*(\boldsymbol{\theta}) + \mathbb{E}_{y \sim \boldsymbol{\eta}}[\Omega_T(\boldsymbol{\rho}(y))] - \langle \boldsymbol{\theta}, \mathbb{E}_{y \sim \boldsymbol{\eta}}[\boldsymbol{\rho}(y)] \rangle$$

is differentiable and convex in $\boldsymbol{\theta}$. If $\boldsymbol{\theta}^* \in \mathbb{R}^d$ is the minimizer of $R_{L_{\Omega_T}}(\cdot, \boldsymbol{\eta})$, its optimality condition indicates

$$\nabla_{\boldsymbol{\theta}} R_{L_{\Omega_T}}(\boldsymbol{\theta}^*, \boldsymbol{\eta}) = \nabla\Omega_T^*(\boldsymbol{\theta}^*) - \mathbb{E}_{y \sim \boldsymbol{\eta}}[\boldsymbol{\rho}(y)] = 0.$$

When $\boldsymbol{\eta} \in \text{relint}(\Delta^K)$, we have $\mathbb{E}_{y \sim \boldsymbol{\eta}}[\boldsymbol{\rho}(y)] \in \text{relint}(\text{conv}(\boldsymbol{\rho}(\mathcal{Y})))$, and thus $\boldsymbol{\theta}^*$ satisfying the above optimality condition does exist. Finally, we conclude the proof by combining Lemmas 8 and 12. ∎

## C  Deferred Proofs in Section 4

### C.1  Proof of Lemma 18

**Proof.** Denote by $V_{\boldsymbol{\theta}}(\boldsymbol{\pi}) := \sum_{i=1}^K e^{\theta_i + 1 - \pi_i}$. Since $\ln(\cdot)$ is strictly increasing on $\mathbb{R}_{>0}$, $V_{\boldsymbol{\theta}}(\boldsymbol{\pi})$ shares the same minimizer as $\ln\left(\sum_{i=1}^K e^{\theta_i + 1 - \pi_i}\right)$. The Lagrangian $\mathcal{F}$ of the minimization problem $V_{\boldsymbol{\theta}}$ for $\boldsymbol{\pi} \in \Delta^K$ is written as follows:

$$\mathcal{F}(\boldsymbol{\pi}, \boldsymbol{\alpha}, \beta) = \sum_{i=1}^K e^{\theta_i + 1 - \pi_i} - \boldsymbol{\alpha}^\top \boldsymbol{\pi} + \beta\left(\sum_{i=1}^K \pi_i - 1\right),$$

where $\alpha_1, \ldots, \alpha_K \geq 0$ and $\beta$ are the Lagrangian multipliers. Since $V_{\boldsymbol{\theta}}(\boldsymbol{\pi})$ is convex and differentiable, and the feasible region $\Delta^K$ is convex, the KKT conditions are necessary and sufficient for the optimality of $\boldsymbol{\pi}^* \in \Pi(\boldsymbol{\theta})$, which requires that there exists $(\boldsymbol{\alpha}^*, \beta^*)$ satisfying

$$-e^{\theta_y + 1 - \pi_y^*} - \alpha_y^* + \beta^* = 0, \qquad\qquad \text{for any } y = 1, \ldots, K, \qquad (20)$$

$$\mathbf{1}^\top \boldsymbol{\pi}^* = 1, \quad \boldsymbol{\pi}^* \geq 0, \quad \boldsymbol{\alpha}^* \geq 0, \qquad\qquad\qquad\qquad (21)$$

$$\alpha_y^* \pi_y^* = 0, \qquad\qquad\qquad\qquad \text{for any } y = 1, \ldots, K. \qquad (22)$$

The conditions (21) and (22) indicates that $\pi_y^* > 0 \implies \alpha_y^* = 0$. Then according to (20), we have

$$\pi_y^* > 0 \implies \pi_y^* = \theta_y - (\ln\beta^* - 1). \qquad\qquad (23)$$

Meanwhile, for $\pi_y^* = 0$, (22) indicates that $\alpha_y^* \geq 0$, which further indicates

$$\pi_y^* = 0 \implies \theta_y = \ln(\beta^* - \alpha_y^*) - 1 \leq \ln \beta^* - 1. \tag{24}$$

Then (23) and (24) can be rewritten as follows:

$$\pi_y^* = \max\{\theta_y - (\ln \beta^* - 1), 0\}. \tag{25}$$

According to (21) and $\ln \beta^* - 1 \in \mathbb{R}$, the KKT conditions can be further simplified into the existence of $\tau^* \in \mathbb{R}$ such that the following conditions simultaneously hold:

$$\begin{cases} \pi_y^* = \max\{\theta_y - \tau^*, 0\}, \\ \sum_{y=1}^{K} \max\{\theta_y - \tau^*, 0\} = 1. \end{cases}$$

Denote by $f(\tau) := \sum_{y=1}^{K} \max\{\theta_y - \tau, 0\}$. Then $f$ is continuous on $\mathbb{R}$, and moreover, it is strictly decreasing on $(-\infty, \theta_{(1)}]$ and equals 0 for $\tau \geq \theta_{(1)}$ because of the following expression:

$$f(\tau) = \begin{cases} \sum_{y=1}^{K}(\theta_y - \tau) & \text{if } \tau \leq \theta_{(K)}, \\ \sum_{i=1}^{k-1}(\theta_{(i)} - \tau) & \text{if } \tau \in [\theta_{(k)}, \theta_{(k-1)}] \text{ for } k = 2, \ldots, K, \\ 0 & \text{if } \tau > \theta_{(1)}. \end{cases}$$

Then $n$ defined in line 2 of Algorithm 1 exists since $f(\theta_{(1)}) = 0 < 1$. We also have that $f(\theta_{(n)}) = \sum_{i=1}^{n}(\theta_{(i)} - \theta_{(n)}) < 1$. When $n < K$, we have that $f(\theta_{(n+1)}) = \sum_{i=1}^{n+1}(\theta_{(i)} - \theta_{(n+1)}) = \sum_{i=1}^{n}(\theta_{(i)} - \theta_{(n+1)}) \geq 1$ by definition. Then we see

$$\frac{\sum_{i=1}^{n} \theta_{(i)} - 1}{n} \in [\theta_{(n+1)}, \theta_{(n)}]$$

and

$$f\left(\frac{\sum_{i=1}^{n} \theta_{(i)} - 1}{n}\right) = \sum_{i=1}^{n}\left(\theta_{(i)} - \frac{\sum_{i=1}^{n} \theta_{(i)} - 1}{n}\right)$$

$$= \sum_{i=1}^{n} \theta_{(i)} - \sum_{i=1}^{n} \theta_{(i)} + 1$$

$$= 1.$$

When $n = K$, we have that $\sum_{i=1}^{K} \theta_{(i)} - K\theta_{(k)} < 1$, that is,

$$\frac{\sum_{i=1}^{K} \theta_{(i)} - 1}{K} < \theta_{[K]}.$$

Then

$$f\left(\frac{\sum_{i=1}^{K} \theta_{(i)} - 1}{K}\right) = \sum_{i=1}^{K}\left(\theta_{(i)} - \frac{\sum_{i=1}^{K} \theta_{(i)} - 1}{K}\right)$$

$$= \sum_{i=1}^{K} \theta_{(i)} - \sum_{i=1}^{K} \theta_{(i)} + 1$$

$$= 1.$$

Combining these two cases, we have

$$\tau^* = \frac{\sum_{i=1}^{n} \theta_{(i)} - 1}{n} < \theta_{(1)},$$

which is what line 3 of Algorithm 1 returns. Since $f(\tau)$ is strictly decreasing on $(-\infty, \theta_{(1)}]$, $\tau^*$ uniquely satisfies $f(\tau^*) = 1$.

Since $\sum_{i=1}^{k} \theta_{(i)}$ can be calculated cumulatively in $\mathcal{O}(K)$ time and the quick sort runs in $\mathcal{O}(K \log K)$ time, Algorithm 1 runs in $\mathcal{O}(K \log K)$ time in total. Furthermore, since $\tau^*$ is unique and $\pi_y^* = \max\{\theta_y - \tau^*, 0\}$, we ensure that $\Pi(\boldsymbol{\theta})$ is a singleton, which concludes the proof. ∎

## C.2 Proof of Proposition 19

**Proof.** By Definition 10, the statement is equivalent to
$$\operatorname*{argmax}_{t \in \mathcal{Y}} \theta_t = \operatorname*{argmax}_{t \in \mathcal{Y}} \pi_{\log}(\boldsymbol{\theta})_t.$$

According to Lemma 18 and Algorithm 1, there exists $\tau \in \mathbb{R}$ such that $\pi_{\log}(\theta)_i = \max\{\theta_i - \tau, 0\}$. We also have that $\max_t \theta_t > \tau$ by contradiction: if $\max_t \theta_t \leq \tau$, $\boldsymbol{\pi}_{\log}(\boldsymbol{\theta}) = \mathbf{0}$ holds, which contradicts $\boldsymbol{\pi}_{\log}(\boldsymbol{\theta}) \in \Delta^K$.

Denote the set $\operatorname{argmax}_{t \in \mathcal{Y}} \theta_t$ by $\mathcal{I}$. For any $i \in \mathcal{I}$, we have $\pi_{\log}(\boldsymbol{\theta})_i = \max\{\theta_i - \tau, 0\} = \max_t \theta_t - \tau > 0$. For any $j \notin \mathcal{I}$,

- If $\theta_j > \tau$: $\pi_{\log}(\boldsymbol{\theta})_j = \max\{\theta_j - \tau, 0\} = \theta_j - \tau < \max_t \theta_t - \tau$,
- If $\theta_j \leq \tau$: $\pi_{\log}(\boldsymbol{\theta})_j = \max\{\theta_j - \tau, 0\} = 0 < \max_t \theta_t - \tau$.

These imply that for any $i \in \mathcal{I}$, $\pi_{\log}(\boldsymbol{\theta})_i \geq \pi_{\log}(\boldsymbol{\theta})_j$ and the equality holds if and only if $j \in \mathcal{I}$, which concludes the proof. ∎

# D  Additional Examples

In this section, we further provide more examples of target losses to demonstrate that we can generate convex smooth surrogate losses for a wide range of target prediction problems. The generated convolutional Fenchel–Young losses are automatically guaranteed to entail linear surrogate regret bounds thanks to Theorem 13.

## D.1  Multiclass Classification with Rejection

**Problem setup.** In multiclass classification with rejection [25], the class space is $\mathcal{Y} = [K]$, and the prediction space $\widehat{\mathcal{Y}} = [K+1]$, which is augmented by a rejection option $K + 1$. We focus on the case that rejection cost $c$ is in $[0, 0.5)$ here. For an input instance with class probability $\boldsymbol{\eta} \in \Delta^K$, the goal is to predict the most likely class label $t \in \operatorname{argmax}_{t \in \mathcal{Y}} \eta_t$ if $\max_{y \in \mathcal{Y}} \eta_y > 1 - c$ for the predetermined cost $c$, and refrain from predicting otherwise. The standard target loss is the 0-1-$c$ loss:
$$\ell_{01c}(t, y) = \begin{cases} [\![ t \neq y ]\!], & t \in [K], \\ c, & t = K+1, \end{cases}$$

that is, the prediction suffers from the ordinary classification error if it is a wrong class label, and suffers from an intermediate error $c$ if it chooses to refrain from prediction.

We adopt the decomposition (3) for this task: $\boldsymbol{\ell}^{\boldsymbol{\rho}}(t) = [\ell_{01c}(t, 1), \cdots, \ell_{01}(t, K)] = \mathbf{1} - \boldsymbol{e}_t$ if $t \neq K + 1$, and $\boldsymbol{\ell}^{\boldsymbol{\rho}}(t) = c\mathbf{1}$ if $t = K + 1$. We choose the label encoding function $\boldsymbol{\rho}(y)$... as in the multiclass classification task. In this case, $N = K + 1$ and $d = K$, and $\mathcal{L}^{\boldsymbol{\rho}} = [\mathbf{1}\mathbf{1}^{\top} - I, c\mathbf{1}]$, where $\mathbf{1}$ is $K$-dimensional.

**Loss formulation and calculation.** In this case, we consider the Shannon negentropy $\Omega$ as in Section 4. Based on the discussion above and Lemma 8, the conjugated convolutional negentropy $\Omega_T^*$ can be written as follows:[6]

$$\Omega_T^*(\boldsymbol{\theta}) = \min_{\boldsymbol{\pi} \in \Delta^{K+1}} \Omega^*(\boldsymbol{\theta} + \mathcal{L}^{\boldsymbol{\rho}}\boldsymbol{\pi}) = \min_{\boldsymbol{\pi} \in \Delta^{K+1}} \ln \left( \sum_{i=1}^{K} \exp(\theta_i + 1 - \pi_i - (1 - c)\pi_{K+1}) \right), \quad (26)$$

and we can then get the corresponding convolutional Fenchel–Young loss as follows, by noting $\Omega_T(\boldsymbol{\rho}(y)) = 0$ for all $y \in \mathcal{Y}$:

$$L_{\Omega_T}(\boldsymbol{\theta}, y) = \min_{\boldsymbol{\pi} \in \Delta^{K+1}} \ln \left( \sum_{i=1}^{K} \exp(\theta_i + 1 - \pi_i - (1 - c)\pi_{K+1}) \right) - \theta_y. \quad (27)$$

To compute $L_{\Omega_T}$, we need to know the minimizer $\boldsymbol{\pi}$ in (26). We show its closed form below.

**Lemma 22** *For any $\boldsymbol{\theta} \in \mathbb{R}^K$, the problem (26) has a unique minimizer $\boldsymbol{\pi}^*(\boldsymbol{\theta}) \in \Pi(\boldsymbol{\theta})$, which can be obtained in $\mathcal{O}(K)$ time. Denote by $y^*$ an arbitrary element in $\operatorname{argmax}_{y' \in [K]} \theta_{y'}$,[7] then the minimizer*

---

[6] We emphasize that the domain of conjugate is $K$-dimensional, despite the $\boldsymbol{\pi}$ is in $K + 1$-simplex.

[7] Though chosen arbitrarily, uniqueness is guaranteed: argmax sets are singleton in the first and third cases.

*can be written as follows:*

$$\boldsymbol{\pi}^*(\boldsymbol{\theta}) = \begin{cases} \boldsymbol{e}_{y^*}, & \text{if } \frac{\exp(\theta_{y^*})}{\exp(\theta_{y^*}) + \sum_{i=1, i\neq y^*}^{K} \exp(\theta_i + 1)} > 1 - c \\ \boldsymbol{e}_{K+1}, & \text{if } \frac{\exp(\theta_{y^*})}{\sum_{i=1}^{K} \exp(\theta_i)} < 1 - c \\ \gamma(\boldsymbol{\theta})\boldsymbol{e}_{y^*} + (1 - \gamma(\boldsymbol{\theta}))\boldsymbol{e}_{K+1}, & \text{else,} \end{cases}$$

*where* $\gamma(\boldsymbol{\theta}) := -\ln\left(\frac{\sum_{i=1}^{K} \exp(\theta_i)}{\exp(\theta_{y^*})} - 1\right) - \ln\left(\frac{1-c}{c}\right).$

**Proof.** First, the Lagrangian of the minimization problem (26) is written as follows:

$$\mathcal{F}(\boldsymbol{\pi}, \boldsymbol{\alpha}, \beta) = \ln\left(\sum_{i=1}^{K} \exp(\theta_i + 1 - \pi_i - (1-c)\pi_{K+1})\right) - \boldsymbol{\alpha}^\top\boldsymbol{\pi} + \beta\left(\sum_{i=1}^{K+1} \pi_i - 1\right)$$

$$= \ln\left(\sum_{i=1}^{K} \exp(\theta_i + 1 - \pi_i)\right) - (1-c)\pi_{K+1} - \boldsymbol{\alpha}^\top\boldsymbol{\pi} + \beta\left(\sum_{i=1}^{K+1} \pi_i - 1\right),$$

where $\alpha_1, \ldots, \alpha_{K+1} \geq 0$ and $\beta \in \mathbb{R}$ are the Lagrangian multipliers. Since the log-sum-exp term is convex and differentiable and the feasible region $\Delta^{K+1}$ is convex, the KKT conditions are necessary and sufficient for the optimality of $\boldsymbol{\pi}^* \in \Pi(\boldsymbol{\theta})$, which requires that there exists $(\boldsymbol{\alpha}^*, \beta^*)$ satisfying

$$\frac{\exp(\theta_y + 1 - \pi_y^*)}{\sum_{i=1}^{K} \exp(\theta_i + 1 - \pi_i^*)} = \beta^* - \alpha_y^*, \qquad \text{for any } y = 1, \ldots, K, \qquad (28)$$

$$1 - c = \beta^* - \alpha_{K+1}^*, \qquad (29)$$

$$\boldsymbol{1}^\top\boldsymbol{\pi}^* = 1, \quad \boldsymbol{\pi}^* \geq 0, \quad \boldsymbol{\alpha}^* \geq 0, \qquad (30)$$

$$\alpha_y^*\pi_y^* = 0, \qquad \text{for any } y = 1, \ldots, K+1. \qquad (31)$$

We continue the proof with the following four observations:

**Observation 1.** By noting $1 - c > 0.5$, we have

$$\frac{\exp(\theta_{y^*})}{\sum_{i=1}^{K} \exp(\theta_i)} \geq 1 - c \implies \underset{y' \in [K]}{\operatorname{argmax}} \, \theta_{y'} = \{y^*\}.$$

**Observation 2.** From the KKT conditions, we have

$$\frac{\exp(\theta_{y^*})}{\exp(\theta_{y^*}) + \sum_{i=1, i\neq y^*}^{K} \exp(\theta_i + 1)} > 1 - c \implies \boldsymbol{\pi}^* = \boldsymbol{e}_{y^*}.$$

To see this, let us integrate the above left-hand side with (28) and (29):

$$\begin{aligned} \beta^* - \alpha_{y^*}^* &= \frac{\exp(\theta_{y^*} + 1 - \pi_{y^*}^*)}{\sum_{i=1}^{K} \exp(\theta_i + 1 - \pi_i^*)} && \text{by (28)} \\ &= \frac{\exp(\theta_{y^*})}{\exp(\theta_{y^*}) + \sum_{i=1, i\neq y^*}^{K} \exp(\theta_i + \pi_{y^*}^* - \pi_i^*)} \\ &> \frac{\exp(\theta_{y^*})}{\exp(\theta_{y^*}) + \sum_{i=1, i\neq y^*}^{K} \exp(\theta_i + 1)} && \text{by } \pi_{y^*}^* \leq 1 \text{ and } \pi_i^* \geq 0 \\ &\overset{(\clubsuit)}{>} 1 - c && \text{by the assumption} \\ &= \beta^* - \alpha_{K+1}^*, && \text{by (29)} \end{aligned}$$

which indicates that $\alpha_{K+1}^* > \alpha_{y^*}^* \geq 0$, and thus $\pi_{K+1}^* = 0$ due to (30) and (31). On the other hand, for any $y \in [K] \setminus \{y^*\}$, we have

$$\beta^* - \alpha_y^* = \frac{\exp(\theta_y + 1 - \pi_y^*)}{\sum_{i=1}^{K} \exp(\theta_i + 1 - \pi_i^*)} \qquad \text{by (28)}$$

$$\leq 1 - \frac{\exp(\theta_{y^*} + 1 - \pi^*_{y^*})}{\sum_{i=1}^{K} \exp(\theta_i + 1 - \pi^*_i)}$$

$$\leq 1 - (1 - c) \qquad \text{by the same argument as } (\clubsuit)$$

$$= c$$

$$< 1 - c \qquad \text{by } c \in [0, 0.5)$$

$$= \beta^* - \alpha^*_{K+1}, \qquad \text{by (29)}$$

and thus $\alpha^*_y > \alpha^*_{K+1} > 0$, which indicates that $\pi^*_y = 0$ except for $y = y^*$ due to (30) and (31). Thus, we verify $\boldsymbol{\pi}^* = \boldsymbol{e}_{y^*}$.

**Observation 3.** From the KKT conditions, we have

$$\frac{\exp(\theta_{y^*})}{\sum_{i=1}^{K} \exp(\theta_i)} < 1 - c \implies \boldsymbol{\pi}^* = \boldsymbol{e}_{K+1}.$$

To see this, we show the contraposition of Observation 3. Suppose there exists $y \in [K]$ such that $\pi^*_y > 0$. Then, we have $\alpha^*_y = 0$ due to (31), and consequently

$$\frac{\exp(\theta_y + 1 - \pi^*_y)}{\sum_{i=1}^{K} \exp(\theta_i + 1 - \pi^*_i)} = \beta^* \qquad \text{by (28) and } \alpha^*_y = 0$$

$$\geq \beta^* - \alpha^*_{K+1} \quad \text{by (30)} \tag{32}$$

$$= 1 - c. \qquad \text{by (29)}$$

Then, for any $y' \in [K] \setminus \{y\}$, this inequality implies

$$\beta^* - \alpha^*_{y'} = \frac{\exp(\theta_{y'} + 1 - \pi^*_{y'})}{\sum_{i=1}^{K} \exp(\theta_i + 1 - \pi^*_i)} \qquad \text{by (28)}$$

$$\leq 1 - \frac{\exp(\theta_y + 1 - \pi^*_y)}{\sum_{i=1}^{K} \exp(\theta_i + 1 - \pi^*_i)}$$

$$\leq 1 - (1 - c) \qquad \text{by (32)}$$

$$= c$$

$$< 1 - c, \qquad \text{by } c \in [0.0.5)$$

which indicates that $\alpha^*_{y'} > 0$ and $\pi^*_{y'} = 0$ due to (30) and (31). Then, we have

$$\frac{\exp(\theta_y + 1 - \pi^*_y)}{\exp(\theta_y + 1 - \pi^*_y) + \sum_{i=1, i \neq y}^{K} \exp(\theta_i + 1)} = \frac{\exp(\theta_y - \pi^*_y)}{\exp(\theta_y - \pi^*_y) + \sum_{i=1, i \neq y}^{K} \exp(\theta_i)}$$

$$= \frac{\exp(\theta_y)}{\exp(\theta_y) + \sum_{i=1, i \neq y}^{K} \exp(\theta_i + \pi^*_y)}$$

$$\geq 1 - c,$$

where we used (32) at the last line. This inequality further implies

$$\frac{\exp(\theta_{y^*})}{\sum_{i=1}^{K} \exp(\theta_i)} \geq \frac{\exp(\theta_y)}{\exp(\theta_y) + \sum_{i=1, i \neq y}^{K} \exp(\theta_i + \pi^*_y)} \geq 1 - c.$$

Thus, the contraposition of Observation 3 is shown.

**Observation 4.** From the KKT conditions again, we have

$$\frac{\exp(\theta_{y^*})}{\sum_{i=1}^{K} \exp(\theta_i)} \geq 1 - c \geq \frac{\exp(\theta_{y^*})}{\exp(\theta_{y^*}) + \sum_{i=1, i \neq y^*}^{K} \exp(\theta_i + 1)}$$

$$\implies \boldsymbol{\pi}^* = \gamma(\boldsymbol{\theta})\boldsymbol{e}_{y^*} + (1 - \gamma(\boldsymbol{\theta}))\boldsymbol{e}_{K+1},$$

where $\gamma(\boldsymbol{\theta}) = -\ln\left(\frac{\sum_{i=1}^{K} \exp(\theta_y)}{\exp(\theta_{y^*})} - 1\right) - \ln\left(\frac{1-c}{c}\right)$. To see this, we carefully examine $\boldsymbol{\pi}^*$. First, if there exists different $y', y'' \in [K]$ with $\pi_{y'}, \pi_{y''} > 0$, we have

$$\beta^* = \frac{\exp(\theta_{y'} + 1 - \pi^*_{y'})}{\sum_{i=1}^{K} \exp(\theta_i + 1 - \pi^*_i)} \qquad \text{by (28) and } \alpha^*_{y'} = 0 \text{ since } \pi_{y'} > 0$$

$$
= \frac{\exp(\theta_{y''} + 1 - \pi^*_{y''})}{\sum_{i=1}^{K} \exp(\theta_i + 1 - \pi^*_i)} \qquad \text{by (28) and } \alpha^*_{y''} = 0 \text{ since } \pi_{y'} > 0
$$
$$
\geq \beta^* - \alpha^*_{K+1} \qquad\qquad\qquad\quad \text{by } \alpha^*_{K+1} \geq 0
$$
$$
= 1 - c \qquad\qquad\qquad\qquad\quad\ \text{by (29)}
$$
$$
> 0.5,
$$

which is impossible since

$$
\frac{\exp(\theta_{y'} + 1 - \pi^*_{y'})}{\sum_{i=1}^{K} \exp(\theta_i + 1 - \pi^*_i)} + \frac{\exp(\theta_{y''} + 1 - \pi^*_{y''})}{\sum_{i=1}^{K} \exp(\theta_i + 1 - \pi^*_i)} \leq 1.
$$

This contradiction indicates that there exists at most one $y \in [K]$ such that $\pi^*_y > 0$. Second, we show $y \in [K]$ with $\pi^*_y > 0$ must be $y = y^*$. To see this, suppose there exists $y \in [K]$ such that $\pi^*_y > 0$ but $y \neq y^*$, then $\pi^*_{y'} = 0$ for any $y' \in [K] \setminus \{y\}$ and we have

$$
\frac{\exp(\theta_y + 1 - \pi^*_y)}{\sum_{i=1}^{K} \exp(\theta_i + 1 - \pi^*_i)} = \frac{\exp(\theta_y - \pi^*_y)}{\exp(\theta_y - \pi^*_y) + \sum_{i=1, i \neq y}^{K} \exp(\theta_i)}
$$
$$
= \beta^* - \alpha^*_y \qquad\qquad\qquad\qquad \text{by (28)}
$$
$$
= \beta^* \qquad\qquad\qquad\qquad\qquad\quad \text{by } \pi^*_y > 0 \text{ and (31)}
$$
$$
\geq \beta^* - \alpha^*_{K+1} \qquad\qquad\qquad\quad \text{by } \alpha^*_{K+1} \geq 0
$$
$$
\geq 1 - c \qquad\qquad\qquad\qquad\qquad \text{by (29)}
$$
$$
> 0.5.
$$

However, this contradicts the following inequality:

$$
\frac{\exp(\theta_{y^*} + 1 - \pi^*_{y^*})}{\sum_{i=1}^{K} \exp(\theta_i + 1 - \pi^*_i)} = \frac{\exp(\theta_{y^*})}{\exp(\theta_y - \pi^*_y) + \sum_{i=1, i \neq y}^{K} \exp(\theta_i)}
$$
$$
\geq \frac{\exp(\theta_{y^*})}{\sum_{i=1}^{K} \exp(\theta_i)} \qquad\qquad \text{by } \pi^*_y \geq 0
$$
$$
\geq 1 - c \qquad\qquad\qquad\quad\ \text{by the assumption}
$$
$$
> 0.5.
$$

Thus, we have verified that $\pi^*_y > 0$ is possible with $y = y^*$ or $y = K + 1$ only. From this, we have

$$
\boldsymbol{\pi}^* = \pi^*_{y^*} \boldsymbol{e}_{y^*} + (1 - \pi^*_{y^*}) \boldsymbol{e}_{K+1}. \tag{33}
$$

From now on, we show that

$$
\begin{cases} \boldsymbol{\pi}^* = \gamma(\boldsymbol{\theta}) \boldsymbol{e}_{y^*} + (1 - \gamma(\boldsymbol{\theta})) \boldsymbol{e}_{K+1} \\ \boldsymbol{\alpha}^* = \boldsymbol{0} \\ \beta^* = 1 - c \end{cases} \tag{34}
$$

fulfill the KKT conditions under the assumption of Observation 4. By using (33), we have

$$
\frac{\exp(\theta_{y^*} + 1 - \pi^*_{y^*})}{\sum_{i=1}^{K} \exp(\theta_i + 1 - \pi^*_i)} = \frac{\exp(\theta_{y^*} - \pi^*_{y^*})}{\exp(\theta_{y^*} - \pi^*_{y^*}) + \sum_{i=1, i \neq y^*}^{K} \exp(\theta_i)} \qquad \text{by (33)}
$$
$$
= \beta^* - \alpha^*_{y^*} \qquad\qquad\qquad \text{by (28)}
$$
$$
= 1 - c + \alpha^*_{y^*} - \alpha^*_{K+1}. \qquad \text{by (29)}
$$

By elementary algebra and plugging in $\boldsymbol{\alpha}^* = \boldsymbol{0}$ given by (34), we can solve it with respect to $\pi^*_{y^*}$ as follows:

$$
\pi^*_{y^*} = -\ln\left( \frac{\sum_{i=1}^{K} \exp(\theta_i)}{\exp(\theta_{y^*})} - 1 \right) - \ln\left( \frac{1 - c}{c} \right) = \gamma(\boldsymbol{\theta}). \tag{35}
$$

By rearranging the assumption of Observation 4, we have

$$\frac{1}{e}\frac{c}{1-c} \leq \frac{\sum_{i=1}^{K} \exp(\theta_i)}{\exp(\theta_{y^*})} - 1 \leq \frac{c}{1-c},$$

from which we can see $0 \leq \pi_{y^*}^* = \gamma(\boldsymbol{\theta}) \leq 1$. All in all, $(\boldsymbol{\pi}^*, \boldsymbol{\alpha}^*, \beta^*)$ in (34) fulfill the KKT conditions, and thus Observation 4 is verified.

By combining Observations 2, 3, and 4, we have shown the closed form of $\pi^*(\boldsymbol{\theta})$. Furthermore, Observation 1 guarantees the uniqueness of $\boldsymbol{\pi}^*(\boldsymbol{\theta})$.

Lastly, it is easy to see that $\boldsymbol{\pi}^*(\boldsymbol{\theta})$ can be computed in linear time because both of the following terms

$$\frac{\exp(\theta_{y^*})}{\exp(\theta_{y^*}) + \sum_{i=1,i\neq y^*}^{K} \exp(\theta_i + 1)} \quad \text{and} \quad \frac{\exp(\theta_{y^*})}{\sum_{i=1}^{K} \exp(\theta_i)}$$

can be computed in linear time. ∎

### D.2 Multilabel Learning with Precision@$k$

**Problem setup.** In multilabel learning, the target prediction space $\mathcal{Y} = [2^d]$ is the collection of indices of all possible combinations of $d$ binary labels with $|\mathcal{Y}| = K = 2^d$. Precision@$k$ is a common performance metric for multilabel ranking. We consider that the prediction space $\widehat{\mathcal{Y}} = [\binom{d}{k}]$ is the collection of indices of all possible size-$k$ subsets of multilabels with $|\widehat{\mathcal{Y}}| = N = \binom{d}{k}$. In addition, we encode the label $y \in \mathcal{Y}$ into $\boldsymbol{\rho}(y) \in \{0,1\}^d$ and the prediction $t \in \widehat{\mathcal{Y}}$ into $\boldsymbol{\mu}(t) \in \{0,1\}^d$, where $\{\boldsymbol{\mu}(t)\}_{t=1}^{N}$ is the collection of all distinct permutations of $\boldsymbol{\omega} = \boldsymbol{e}_1 + \cdots + \boldsymbol{e}_k$. Then, the target loss of Precision@$k$ is defined as follows:

$$\ell(t, y) = 1 - \frac{\sum_{i=1}^{d} \rho(y)_i \mu(t)_i}{k}.$$

This is the portion of binary labels with value 0 in the top-$k$ list.

We adopt the decomposition (3) by using the aforementioned $\boldsymbol{\rho}(y)$, $\boldsymbol{\ell}^\rho(t) = -\frac{\boldsymbol{\mu}(t)}{k}$, and $c(y) = 1$ for all $y \in \mathcal{Y}$.

**Loss formulation and calculation.** For Precision@$k$, we consider the base negentropy

$$\Omega(\boldsymbol{p}) = \sum_{i=1}^{d} (p_i \ln p_i + (1 - p_i) \ln(1 - p_i)).$$

Then, we have

$$\Omega^*(\boldsymbol{\theta}) = \sum_{i=1}^{d} \ln(1 + \exp(\theta_i)),$$

and

$$\Omega_T^*(\boldsymbol{\theta}) = \min_{\boldsymbol{\pi} \in \Delta^{\binom{d}{k}}} \sum_{i=1}^{d} \ln\left(1 + \exp\left(\theta_i - \frac{1}{k}\sum_{t=1}^{\binom{d}{k}} \pi_t \mu(t)_i\right)\right). \tag{36}$$

We can simplify it into the following problem:

$$\Omega_T^*(\boldsymbol{\theta}) = \min_{\boldsymbol{v} \in V} \sum_{i=1}^{d} \ln\left(1 + \exp\left(\theta_i - \frac{1}{k}v_i\right)\right), \tag{37}$$

where $V := \{\boldsymbol{v} \in \mathbb{R}^d : v_i \in [0,1], \|\boldsymbol{v}\|_1 = k\}$. Then, the convolutional Fenchel–Young loss can be written as follows:

$$L_{\Omega_T}(\boldsymbol{\theta}, y) = \min_{\boldsymbol{v} \in V} \sum_{i=1}^{d} \ln\left(1 + \exp\left(\theta_i - \frac{1}{k}v_i\right)\right) - \langle \boldsymbol{\rho}(y), \boldsymbol{\theta}\rangle.$$

To calculate the gradient of $L_{\Omega_T}(\cdot, y)$, we only need the solution $\boldsymbol{v}^*$ in the optimization problem (37), which can be obtained efficiently.

---

**Algorithm 2** Exact Solution of (37) in $\mathcal{O}(d \ln d)$

---

1: Set $f(\lambda) = \sum_{i=1}^{d} \max\{0, \min\{\theta_i - \lambda, 1\}\}$
2: Sort $\tilde{\boldsymbol{\theta}} = [\boldsymbol{\theta}; \boldsymbol{\theta} - \mathbf{1}] \in \mathbb{R}^{2d}$ such that $\tilde{\theta}_{(1)} \geq \cdots \geq \tilde{\theta}_{(2d)}$.
3: $n \leftarrow \max\{n \in [2d] : f(\tilde{\theta}_{(n)}) < k\}$.
4: $\lambda^* \leftarrow \tilde{\theta}_{(n)} + \frac{k - f(\tilde{\theta}_{(n)})}{f(\tilde{\theta}_{(n+1)}) - f(\tilde{\theta}_{(n)})}(\tilde{\theta}_{(n+1)} - \tilde{\theta}_{(n)})$.
5: $v_i^* \leftarrow \max\{0, \min\{\theta_i - \lambda^*, 1\}\}$.

---

**Lemma 23** *For any $\boldsymbol{\theta} \in \mathbb{R}^d$, the minimization problem* (37) *has a unique minimizer $\boldsymbol{v}^*$, which can be obtained in $\mathcal{O}(d \log d)$ time with Algorithm 2.*

**Proof.** The uniqueness can be seen by noting that the objective function is strictly convex and its domain $V$ is compact and convex.

Let us analyze the Lagrangian of the minimization problem (37), which can be written as follows:

$$\mathcal{F}(\boldsymbol{v}, \boldsymbol{\alpha}, \boldsymbol{\beta}, \gamma) = \sum_{i=1}^{d} \ln\left(1 + \exp\left(\theta_i - \frac{1}{k}v_i\right)\right) - \boldsymbol{\alpha}^\top \boldsymbol{v} + \boldsymbol{\beta}^\top(\boldsymbol{v} - \mathbf{1}) + \gamma\left(\sum_{i=1}^{d} v_i - k\right), \quad (38)$$

where $\boldsymbol{\alpha} \in \mathbb{R}_{\geq 0}^d$, $\boldsymbol{\beta} \in \mathbb{R}_{\geq 0}^d$, and $\gamma \in \mathbb{R}$ are the Lagrangian multipliers. Then, we have the following KKT conditions:

$$\frac{\exp(\theta_i - v_i^*)}{1 + \exp(\theta_i - v_i^*)} = \gamma^* + \beta_i^* - \alpha_i^*, \qquad \text{for any } i = 1, \ldots, d \qquad (39)$$

$$\mathbf{1}^\top \boldsymbol{v}^* = k, \quad \mathbf{0} \leq \boldsymbol{v}^* \leq \mathbf{1}, \quad \boldsymbol{\alpha}^* \geq \mathbf{0}, \quad \boldsymbol{\beta}^* \geq \mathbf{0}, \qquad (40)$$

$$\alpha_i^* v_i^* = 0, \quad \beta_i^*(1 - v_i^*) = 0, \qquad \text{for any } i = 1, \ldots, d \qquad (41)$$

where the inequalities in (40) are element-wise.

First, we show that the KKT conditions are fulfilled by $v_i^* = \max\{0, \min\{\theta_i - \lambda^*, 1\}\}$ for each $i \in [d]$, where $\lambda^* := \ln(\gamma^*/(1 - \gamma^*))$. Fixing $i \in [d]$, we divide into cases. If $v_i^* > \theta_i - \lambda^*$, we have

$$\frac{\exp(\theta_i - v_i^*)}{1 + \exp(\theta_i - v_i^*)} < \gamma^*,$$

which implies $\beta_i^* - \alpha_i^* < 0$ by (39). Since $\beta_i^* > 0$ by (40), we have $\alpha_i^* > 0$, which implies $v_i^* = 0$ due to (41). If $v_i^* < \theta_i - \lambda^*$, we can show $v_i^* = 1$ similarly. Thus, we have

- If $v_i^* > \theta_i - \lambda^*$, then $v_i^* = 0$;
- If $v_i^* < \theta_i - \lambda^*$, then $v_i^* = 1$.

Moreover, let us divide into cases on $\theta_i - \lambda^*$. If $\theta_i - \lambda^* < 0$, we must have $v_i^* > \theta_i - \lambda^*$ since $v_i \in [0, 1]$ (due to the feasibility (40)), which yields $v_i^* = 0$. If $\theta_i - \lambda^* > 1$, we have $v_i^* = 1$ similarly. If $0 \leq \theta_i - \lambda^* \leq 1$, we have $v_i^* = \theta_i - \lambda^*$ otherwise it ends up with contradiction—say, supposing $v_i^* > \theta_i - \lambda^*$, then we have $0 = v_i^* > \theta_i - \lambda^*$, which contradicts the premise $\theta_i - \lambda^* \in [0, 1]$. Combining all above, we have verified that

$$v_i^* = \max\{0, \min\{\theta_i - \lambda^*, 1\}\} \quad \text{for } i \in [d]$$

fulfills the KKT conditions.

Next, we show that Algorithm 2 returns this $\boldsymbol{v}^*$. By noting the constraint $\sum_{i=1}^{d} v_i^* = k$ in the feasibility conditions (40), we have

$$(f(\lambda^*) =) \sum_{i=1}^{d} \max\{0, \min\{\theta_i - \lambda^*, 1\}\} = k.$$

Thus, we need to solve $f(\lambda^*) = k$ to obtain $\boldsymbol{v}^*$. Let us sort the elements of $\tilde{\boldsymbol{\theta}} := [\boldsymbol{\theta}; \boldsymbol{\theta} - \mathbf{1}] \in \mathbb{R}^{2d}$ such that $\tilde{\theta}_{(1)} \geq \cdots \geq \tilde{\theta}_{[2d]}$. The solution to $f(\lambda^*) = k$ uniquely exists since we have

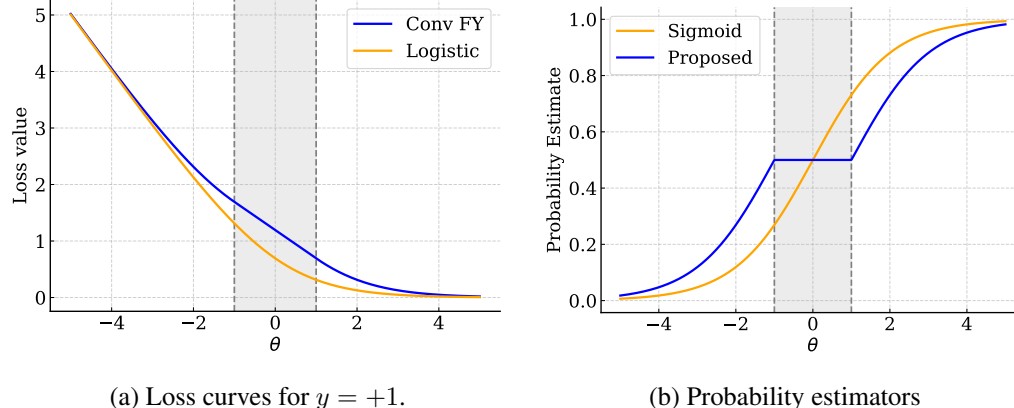

(a) Loss curves for $y = +1$.

(b) Probability estimators

**Figure 1:** Visualization of the convolutional Fenchel–Young loss, logistic loss, and probability estimators.

$f(\tilde{\theta}_{(1)}) = 0 < k$, $f(\tilde{\theta}_{[2d]}) = d > k$, and $f$ is continuous and strictly decreasing on $[\tilde{\theta}_{[2d]}, \tilde{\theta}_{(1)}]$. From the strict decreasing nature, it can also be inferred that $n$ in Step 3 of Algorithm 2 is the largest possible $n \in [2d]$ that $f(\tilde{\theta}_{(n)}) < k$ and $f(\tilde{\theta}_{(n+1)}) \geq k$, which indicates that the solution is in $(\tilde{\theta}_{n+1}, \tilde{\theta}_{(n)}]$. Furthermore, $f$ is linear on each segment $[\tilde{\theta}_{[i+1]}, \tilde{\theta}_{(i)}]$. This indicates that for the point $\lambda^* \in [\tilde{\theta}_{(n+1)}, \tilde{\theta}_{(n)}]$, we have

$$f(\tilde{\theta}_{(n+1)}) + (\lambda^* - \tilde{\theta}_{(n)}) \frac{f(\tilde{\theta}_{(n+1)}) - f(\tilde{\theta}_{(n)})}{\tilde{\theta}_{(n+1)} - \tilde{\theta}_{(n)}} = f(\lambda^*) = k,$$

which yields Step 4 in Algorithm 2.

The time complexity of sorting $\tilde{\theta}$ is $\mathcal{O}(d \log d)$. Note that the computation of $f(z)$ is in $\mathcal{O}(d)$ and $n$ can be found using binary search due to monotonicity of $f$ in $\mathcal{O}(\log d)$ steps, we can conclude that Step 3 is in $\mathcal{O}(d \log d)$. ∎

# E  A Special Case: Visualization on Binary Classification

In this section, we visualize the graphs of convolutional Fenchel–Young losses and the corresponding probability estimator (provided in Theorem 17) for binary classification to get more intuition. For binary classification, we adopt the class space $\mathcal{Y} = \widehat{\mathcal{Y}} = \{-1, +1\}$, and the target loss $\ell_{01}(y, t) = [\![y \neq t]\!]$. We use the following decomposition of the target loss:

$$\rho(+1) = \frac{1}{2}, \ \rho(-1) = -\frac{1}{2}, \ \ell^\rho(+1) = -1, \ \ell^\rho(-1) = 1, \ \text{and} \ c(+1) = c(-1) = \frac{1}{2}.$$

With this decomposition, surrogate losses can operate on a univariate prediction, which is convenient for the visualization purpose.

We compare convolutional Fenchel–Young losses with the standard Fenchel–Young losses generated by the binary Shannon negentropy:

$$\Omega(p) = \left(\frac{1}{2} + p\right) \ln\left(\frac{1}{2} + p\right) + \left(\frac{1}{2} - p\right) \ln\left(\frac{1}{2} - p\right),$$

where $\frac{1}{2} + p =: \eta_{+1}$ is the positive class probability and $\frac{1}{2} - p =: \eta_{-1}$ is the negative class probability. The corresponding convolutional negentropy is as follows:

$$\Omega_T(p) = \left(\frac{1}{2} + p\right) \ln\left(\frac{1}{2} + p\right) + \left(\frac{1}{2} - p\right) \ln\left(\frac{1}{2} - p\right) + \max\{p, -p\},$$

where $\max\{p, -p\} = \frac{1}{2} - \min\{\eta_{+1}, \eta_{-1}\}$ is the negative Bayes 0-1 risk of class probability $(\eta_{+1}, \eta_{-1}) = (\frac{1}{2} + p, \frac{1}{2} - p)$. Then, the conjugate of convolutional negentropy can be written

as follows:

$$\Omega_T^*(\theta) = \begin{cases} \ln(1 + \exp(\theta + 1)) - \frac{\theta+1}{2}, & \text{if } \theta < -1 \\ \ln(2), & \text{if } -1 \leq \theta \leq 1 \\ \ln(1 + \exp(\theta - 1)) - \frac{\theta-1}{2}. & \text{if } 1 < \theta \end{cases}$$

Correspondingly, the convolutional Fenchel–Young loss is

$$L_{\Omega_T}(\theta, y) = \begin{cases} \ln(1 + \exp(1 + \theta)) - \frac{\theta(1+y)+1}{2}, & \text{if } \theta < -1 \\ \ln(2) - \frac{\theta y}{2}, & \text{if } -1 \leq \theta \leq 1 \\ \ln(1 + \exp(\theta - 1)) - \frac{\theta(1+y)-1}{2}. & \text{if } 1 < \theta \end{cases}$$

We can explicitly write down the gradient of the conjugated entropy as follows:

$$\nabla_\theta \Omega_T^*(\theta) = \begin{cases} \frac{\exp(1+\theta)}{1+\exp(1+\theta)} - \frac{1}{2}, & \theta < -1 \\ \frac{\exp(\theta-1)}{1+\exp(\theta-1)} - \frac{1}{2}, & \theta > 1 \\ 0, & \theta \in [-1, 1] \end{cases}$$

which is the estimator of $\eta_{+1}\rho(+1) + \eta_{-1}\rho(-1) = \frac{\eta_{+1}-\eta_{-1}}{2} = \frac{\eta_{+1}-1+\eta_{+1}}{2} = \eta_{+1} - \frac{1}{2}$ by Theorem 17. Finally, we can use the link function $\nabla_\theta \Omega_T^*(\theta) + \frac{1}{2}$ as the estimator of $\eta_{+1}$.

We show the convolutional Fenchel–Young loss and logistic loss (which is the standard Fenchel–Young loss generated by the binary Shannon negentropy) in Figure 1. It can be seen that the convolutional Fenchel–Young loss is linear in the shaded region $\theta \in [-1, 1]$, while resembling the logistic loss outside of this region. Compared with the sigmoid function used for probability estimation with logistic loss, the link function induced by the convolutional Fenchel–Young loss also generates a valid probability estimate in $[0, 1]$ for any $\theta \in \mathbb{R}$, with $\theta \in [-1, 1]$ generates constant value $0.5$ as the estimate.

**Further discussion.** In case of binary classification, Figure 1 (a) nicely illustrates that the convolutional Fenchel–Young loss linearly "extends" the logistic loss at $\theta = 0$, which is the boundary point for binary classification. This illustration is possible because the above formulation for binary classification operates on the univariate score $\theta \in \mathbb{R}$. The linear extension at the boundary point aligns with Frongillo and Waggoner [39], which shows the square-root regret lower bound by assuming that a surrogate loss is locally strongly convex around the boundary points. In general prediction tasks, it is not straightforward to overcome the square-root regret lower bound by such a linear extension because the boundary points for a high-dimensional prediction task can be infinitely many. By contrast, convolutional Fenchel–Young losses provide a general recipe to get linear surrogate regret bound via infimal convolution.

## F  Additional Empirical Results

### F.1  Multiclass Classification

To provide empirical validation of our proposed results, we use the loss introduced in Section 4 as an example and evaluate its performance on the ImageNet-1k dataset [32] using the ResNet-50 architecture [43].

**Experimental Setup.** We follow the default configuration of the PyTorch [71] ImageNet training script. Specifically, we use stochastic gradient descent (SGD) with a momentum of $0.9$, training for 120 epochs with a mini-batch size of 256. The initial learning rate is set to $0.1$ and divided by 10 every 30 epochs. For comparison, we also report results obtained using the standard cross-entropy loss under the same configuration. All experiments are conducted on 8 GeForce RTX 3090 GPUs, and we report the average validation accuracy over 3 independent runs.

**Results and Discussion.** As shown in Table 1, our proposed loss slightly outperforms the standard cross-entropy loss in terms of validation accuracy under a 5% t-test,. This improvement is achieved without any additional hyperparameter tuning, demonstrating the potential of our approach. Meanwhile, the computation time per epoch remains comparable between the two methods, which indicates the efficiency of the proposed loss. These results confirm the compatibility of our loss with both mini-batch and distributed optimization settings.

**Table 1:** Comparison between cross-entropy loss and the proposed loss on ImageNet-1k using ResNet-50.

| Metric | Cross-Entropy Loss | Proposed Loss (18) |
|---|---|---|
| Accuracy (%) | 76.40 | **76.81** |
| Averaged Running Time / Epoch (s) | 647.22 | 653.63 |

## F.2 Classification with Rejection

We further evaluate the proposed loss (27) under the classification with rejection (CwR) setting, where the rejection cost is fixed at $c = 0.05$. The experiments are conducted on the CIFAR-10 and CIFAR-100 datasets [46]. We adopt the WideResNet-28 architecture [96] with a widen factor of 4 and a hidden dimension of 50 as the backbone network for all experiments.

**Experimental Setup.** We train each model for 120 epochs on 8 GeForce RTX 3090 GPUs with a per-GPU batch size of 128. The optimizer is SGD with a momentum of 0.9, weight decay of $5e - 4$, and an initial learning rate of 0.1. The learning rate is scheduled by cosine annealing.

For data augmentation, each image is first converted to a tensor, then padded by 4 pixels on each side using reflection padding, followed by random cropping to $32 \times 32$ and random horizontal flipping. The images are finally normalized using the dataset-specific mean and standard deviation. We report the average system accuracy (1-averaged 0-1-c loss) and acceptance rate over 5 runs. Meanwhile, we also provide the result of ordinary classification using the similar loss (18) we proposed.

**Table 2:** CwR/Classification performance of proposed losses on CIFAR-10/100 using WideResNet-28.

| Metric | CIFAR-10 | CIFAR-100 |
|---|---|---|
| CwR with (27) | | |
| System Accuracy: c=0.05 (%) | 96.79 | 91.90 |
| Acceptance Rate (%) | 92.84 | 65.94 |
| Averaged Running Time / Epoch (s) | 5.79 | 6.64 |
| Classification with (18) | | |
| Accuracy (%) | 93.87 | 74.36 |
| Averaged Running Time / Epoch (s) | 6.07 | 7.41 |

**Results and Discussion.** As shown in Table 2, both CIFAR-10 and CIFAR-100 experiments demonstrate that our proposed loss successfully performs classification with rejection, achieving higher system accuracy while maintaining a reasonable acceptance rate. This indicates that the model learns to reject uncertain samples effectively without sacrificing overall performance. In addition, the average computation time per epoch is even lower than that of ordinary classification, which we attribute to the $\mathcal{O}(K)$ complexity of gradient computation for the proposed formulation, compared with the $\mathcal{O}(K \log K)$ cost in standard classification. These results confirm the efficiency and practicality of our loss for reliable decision-making under uncertainty.

