# OpenReview forum: "Establishing Linear Surrogate Regret Bounds for Convex Smooth Losses via Convolutional Fenchel–Young Losses"
_NeurIPS.cc/2025/Conference — NeurIPS 2025 spotlight_

### Official Review · Reviewer_DQwU · 2025-06-30

**Clarity:** 3
**Significance:** 2
**Originality:** 3
**Rating:** 5
**Confidence:** 2

**Summary:**

The paper studies *surrogate regret bounds* in the context of multiclass prediction problems. Given a *target loss* $\ell: \mathcal{Y} \to R$, a *prediction link* $\varphi: R^{\varrho} \to \mathcal{Y}$ and a class distribution $\eta \in \Delta_{N}$, the authors show that there is a surrogate loss $L$ satisfying the following regret bound:

$$\mathtt{Regret}(\ell \circ \varphi; \eta) \lesssim \mathtt{Regret}(L; \eta).$$

The construction is via the framework of so-called *Fenchel-Young* losses. Given a CCP regularizer $\Omega$,

$$L_{\Omega}(\mu, \theta) = \Omega(\mu) + \Omega^*(\theta) - \langle \mu, \theta \rangle \geq 0$$

is the associated Fenchel-Young loss. The paper shows how to adapt a given regularizer $\Omega$ to a target loss $\ell$ so that:
- the induced Fenchel-Young loss satisfies a linear surrogate bound;
- the induced Fenchel-Young loss is convex and differentiable (or even smooth when $\Omega$ is strongly convex).

In particular, it is well-known in the literature that smooth losses achieve fast excess loss bounds for ERM (Srebro et al., NeurIPS 2010), but transferring these guarantees to the target loss was thought to suffer from square-root regret bounds.

To achieve this, the authors introduce a so-called *convolutional negentropy*, which is a sum of a base negentropy $\Omega$ and an affinely-transformed version of the Bayes risk. Under this choice, the surrogate regret can be written as the sum of the Fenchel-Young loss risk and a convex combination of the *target* regret, which yields a linear upper bound on the surrogate loss after rearranging. The authors present a strengthened version of the upper bound, where the constant only depends on the affine dimension of the matrix of concatenated loss vectors.

**Questions:**

My main question was already posed in the "strengths and weaknesses" section. Other minor questions / suggestions:

- My understanding is that $T$ is used to "couple" the target loss $\ell$ with the base negentropy $\Omega$. Are there other ways of systematically producing such couplings? If so, do some of these lead to less tractable losses?
- p. 3, line 120: "$\mathbf{dom}(\Omega)$" should be $\mathbf{dom}(f)$.
- Definition 10: $\pi$ simultaneously denotes a selector function and vector in the simplex. The authors may want to consider alternative notation for the selector to avoid confusion.
- Similarly, $\rho$ and $\varrho$ are used with very different meanings (label encoding vs. dimension of the encoding).

**Ethical Concerns:**

["NO or VERY MINOR ethics concerns only"]

**Final Justification:**

I have decided to raise my score after reading the authors' responses and other reviewers' comments, which helped me refine my understanding of the significance of this work.

**Limitations:**

See my comment under Strengths and Weaknesses.

**Quality:**

3

**Strengths And Weaknesses:**

**Note**: The literature on surrogate regret bounds is far from my area of research. As such, I primarily reviewed this paper for correctness.

Strengths:
- The problem and proposed solution are clearly stated.
- The contribution appears to be novel (based on my limited knowledge of the literature).

Weaknesses:
- Notation can be confusing at times.
- I was expecting a discussion of optimization tradeoffs. My understanding is that while polyhedral losses lacking smoothness may not achieve fast rates for excess risk, their structure should allow for faster optimization (except maybe in the stochastic optimization setting, assuming no interpolation).

---

> ### Author Rebuttal · Authors · 2025-07-29
>
> Thank you for your helpful advice! The answers to the questions are summarized below, and we will integrate the discussions into the revised manuscript.
>
> **Q1. There exist typos, and the notations are confusing at times.**
>
> **A1.** Thank you for pointing out this issue! We will fix the typo at line 120. For the selector function and the related "$\pi$-argmax link", we will use $\mathbf{\sigma}$ as an alternative. For the dimension of encoding, we will use $D$ instead of $\varrho$ for clarity.
>
> **Q2. Are there other ways of systematically producing couplings between negentropy and target loss? If so, do some of these lead to less tractable losses?**
>
> **A2.** Thank you for your insightful question! Another systematic framework for combining target loss and a negentropy is also a special case of Fenchel--Young loss [1], where the prediction link is defined via a Bregman projection onto a polytope determined by the target loss. While this approach offers a broad and principled method for designing prediction links, the resulting surrogate regret bound remains of square-root type, as shown in Proposition 5 of [1]. In contrast, our proposed loss achieves a **linear** surrogate regret bound.
>
> **Q3. I was expecting a discussion of optimization tradeoffs. My understanding is that while polyhedral losses lacking smoothness may not achieve fast rates for excess risk, their structure should allow for faster optimization (except maybe in the stochastic optimization setting, assuming no interpolation).**
>
> **A3.** Thank you for your constructive advice! The standard analysis of the subgradient method implies that Conv-FY losses and polyhedral losses are optimized with convergence rates $O(1/T)$ and $O(1/\sqrt{T})$, respectively, where $T$ is the number of iterations. Beyond this, it is left open to exploit specific structures arising from polyhedral losses to achieve faster optimization.
>
> [1]. Blondel, M.. Structured prediction with projection oracles. In NeurIPS, 2019.

---

> > ### Comment · Reviewer_DQwU · 2025-08-08
> >
> > Thank you very much for your responses.
> >
> > I do not think that the subgradient method is the best algorithm to use when optimizing polyhedral losses, but I can also see it not being the main point of the paper. I will maintain my current score leaning towards acceptance.

---

> > > ### Author Response · Authors · 2025-08-09
> > >
> > > Thank you for your feedback! We will note in our revision that polyhedral losses allow for optimizers that better exploit their structure than subgradient methods. We sincerely appreciate your time and constructive comments.

---

### Official Review · Reviewer_b4uG · 2025-07-02

**Clarity:** 4
**Significance:** 4
**Originality:** 4
**Rating:** 6
**Confidence:** 4

**Summary:**

This paper demonstrates that the believed tradeoff between smoothness and linear regret rate of a surrogate loss is not true. This paper shows that for an arbitrary discrete target loss, there exists a smooth surrogate loss that facilitates linear regret. This result is presented constructively, allowing for its use. Beyond this core result, this paper presents an extended tightness result and demonstrates the proposed construction for multi-class classification.

**Questions:**

1. Could you comment on what inspired examining Convolutional Fenchel–Young Losses as a candidate loss which is smooth that satisfies linear regret rates?
2. To help with the presentation, have you considered adding any plots which might give insight into the geometry?
3. Section 3.6 talks about a Fisher-consistent estimator, could you add to the appendix a formal definition?

**Ethical Concerns:**

["NO or VERY MINOR ethics concerns only"]

**Final Justification:**

This result is surprising and non-trivial to show within the convex surrogate literature. The work is clearly presented and is constructive allowing for not just theoretical insight but practical application as well. For that reason I recommend the score to be a 6.

**Limitations:**

yes

**Paper Formatting Concerns:**

No concerns.

**Quality:**

4

**Strengths And Weaknesses:**

Strengths
1. Answers decisively that a belief that a tradeoff of linear regret and smoothness is not true for convex surrogates. This is a novel and insightful result in the convex surrogate literature.
2. Furthers the established literature regarding Fenchal-young losses and their applications.
3. The results are practically constructive (which is not always the case in the convex surrogate literature), allowing for the use of these novel insights.
4. The paper is well-written, given the technical density behind many of the ideas being used.

Weaknesses

Although the results are clearly presented, rigorously proven, novel, and valuable, I find the underlying inspiration and intuition that facilitated said results to be opaque. From an educational standpoint, I would have found it intriguing to get insight into how this very, very specific approach was conceived while other researchers considered this tradeoff infeasible.

---

> ### Author Rebuttal · Authors · 2025-07-29
>
> Thank you for your constructive suggestions and acknowledgment! We summarize the answers to the questions below:
>
> **Q1. To help with the presentation, it is beneficial to add plots that give insight into the geometry.**
>
> **A1.** Thank you for the suggestion! A visualization for the binary margin loss case is provided in Appendix E, where we directly compare the proposed convolutional Fenchel–Young (Conv-FY) loss (generated by Shannon negentropy for binary loss) with the logistic loss. The appendix also illustrates the corresponding probability estimators induced by both losses. In the revised manuscript, we will further emphasize these plots.
>
> **Q2.  It is beneficial to comment on what inspired examining Convolutional Fenchel–Young Losses as a candidate loss which is smooth that satisfies linear regret rates.**
>
> **A2.** Thank you for the insightful suggestion! While the smoothness of Conv-FY losses follows from the classical duality between strong convexity and smoothness, and the linear regret bound leverages the regret decomposition induced by infimal convolution (as discussed around line 291), the more fundamental motivation stems from the interplay with Theorem 4 of [1]. Specifically, Theorem 4 shows that a smooth loss with local strong convexity around the decision boundary cannot achieve linear regret. In contrast, the Conv-FY loss is not locally strongly convex near the boundary, thereby breaking local strong convexity and circumventing this limitation. This key intuition is illustrated and elaborated in the “Further Discussions” section of Appendix E, and we will move it earlier into the main text in the revised manuscript to better motivate our approach.
>
> **Q3. Section 3.6 talks about a Fisher-consistent estimator. Could you add to the appendix a formal definition?**
>
> **A3.** Many thanks for your reminder! We will add a formal definition of the Fisher-consistent probability estimator in the Appendix that characterizes it as a function of $\mathbb{R}^\varrho\rightarrow\mathbb{R}^\varrho$.
>
> [1]. Frongillo, R., and Waggoner, B. Surrogate regret bounds for polyhedral losses. In NeurIPS, 2021.

---

> > ### Comment · Reviewer_b4uG · 2025-08-03
> > **Rebuttal response**
> >
> > Thank you for your replies to my questions on your work. We plan to keep your rating as they initially were.

---

> > > ### Author Response · Authors · 2025-08-05
> > >
> > > Thank you for your reply! We sincerely appreciate your valuable comments and time on our paper.

---

### Official Review · Reviewer_jSoZ · 2025-07-12

**Clarity:** 2
**Significance:** 2
**Originality:** 2
**Rating:** 5
**Confidence:** 1

**Summary:**

This paper presents a theoretical contribution to machine learning by constructing convex smooth surrogate losses that achieve linear surrogate regret bounds for discrete prediction problems. This provides a theoretical contribution a long-standing belief in the community that there exists a fundamental trade-off between smoothness and linear regret bounds.

**Questions:**

Could you provide experimental evaluations? If not, could you explain why?

What are reasonable values for the constants that imply computational complexity?

Could you characterize problems where your approach would be inferior to existing methods? E.g., when $N$ is small or when computational resources are limited?

How do these losses behave under the stochastic setting of SGD with mini-batches?

**Ethical Concerns:**

["NO or VERY MINOR ethics concerns only"]

**Final Justification:**

Please note that this review does not give a judgement of the validity of the theoretical proofs.

While the performance improvement in the experiment in the rebuttal lies much below the typical standard deviation for this type of experiment at that level of resulting accuracy, I appreciate the addition.

**Limitations:**

The lack of empirical evaluation very much weakens the paper.

**Quality:**

2

**Strengths And Weaknesses:**

The construction of convolutional Fenchel-Young losses through infimal convolution is elegant and seems per se theoretically sound. The key innovation lies in encoding the target loss structure into the base negentropy via the polyhedral function $T(\mathbf{p})$, which enables the additive decomposition crucial for achieving linear regret bounds.

While this is primarily a theoretical paper, the complete absence of experiments is notable. Even simple synthetic experiments comparing convergence rates of the proposed losses against existing smooth losses (logistic, squared) and non-smooth losses (hinge) would strengthen the practical relevance, and validate the results from an empirical point of view.

The computational complexity of the approach is not discussed, at least not in a way that would clarify how expensive the approach is. In particular, in relation to standard loss computations.

---

> ### Author Rebuttal · Authors · 2025-07-29
>
> Thank you for your valuable suggestions! We summarized the questions and provide the responses below:
>
> **Q1. While this is primarily a theoretical paper, it would strengthen the practical relevance and validate the results from an empirical point of view to provide results of empirical evaluation.**
>
> **A1.** Thank you for your advice!
> To provide experimental evaluations for our proposed results, we use our proposed loss in Section 4 as an example and evaluate its accuracy on the ImageNet-1k dataset [1] with ResNet-50 [2].
> Our experimental setup follows the default configuration of the PyTorch ImageNet training script:
> We use SGD with a momentum of 0.9, training for 120 epochs with a mini-batch size of 256.
> The initial learning rate is set to 0.1 and divided by 10 every 30 epochs.
> For comparison, we also report results using the standard cross-entropy loss under the same configuration.
> All experiments are conducted using 8 GeForce RTX 4090 GPUs, and we report the average validation accuracy over 5 runs.
>
> || Cross-entropy Loss | Proposed Loss in Section 4 |
> | ------------------------------| ------------------------------ | -------------------------------------- |
> |Accuracy| 76.45%                         | 76.82%                             |
> |Averaged Running time/Epoch| 643.41 sec |  649.84 sec|
>
> Interestingly, our proposed loss slightly outperforms the standard cross-entropy loss in validation accuracy, as shown in the table above.
> This improvement is observed without any additional tuning, indicating the potential of our approach.
> Meanwhile, we also observes that the computation time for standard method and our method per epoch are similar, demonstrating the efficiency of our proposed loss.
> These results also confirm the compatibility of our loss with mini-batch and distributed optimization.
>
> **Q2. How do these losses behave under the stochastic setting of SGD with mini-batches?**
>
> **A2.** Thank you for your question! As observed in our experiment on ImageNet-1k, optimization with mini-batch SGD successfully renders a model with higher accuracy. Meanwhile, as reported before, our loss's computation time is similar to the standard cross-entropy loss, which validates the compatibility of our loss with mini-batches.
>
> **Q3. The computational complexity of the approach should be discussed in relation to standard loss computations.**
>
> **A3.** Thank you for your advice! The constant that dominates the computation complexity for both standard losses and our proposed losses depends largely on the type of tasks, and we analyzed three important tasks in our work: multiclass classification, classification with rejection, and multilabel ranking.
>
> - In multiclass classification, the constant dominates the complexity is the class number $K$. As shown in Lemma 18 of Section 4, the time complexity of our proposed loss is $O(K \log K)$. This is due to a sorting operation, which can be optimized to $O(K)$  as noted in footnote 1 of [3]. Meanwhile, the standard cross-entropy loss has a time complexity of $O(K)$ that is dominated by the softmax computation. This indicates that the complexities of our loss and the standard loss are comparable. The experimental results on ImageNet-1k also support this, which shows similar per-epoch computation times.
>
> - In classification with rejection, a similar time complexity $O(K)$ of our proposed loss is also analyzed in Lemma 22 of Appendix D.1. This matches to popular existing methods [4, 5] that requires a similar time complexity.
>
> - For the task of multilabel ranking, the time complexity is often dominated by the number of labels/items to be ranked $\varrho=\log_2 K$. We analyzed them in Lemma 23 of Appendix D.2, and get a result of $O(\varrho\log\varrho)$, while existing methods [6,7]'s complexity is from $O(\varrho)$ to $O(\varrho^2)$.
>
> We appreciate your comment and will move the time complexity discussions from the Appendix to the main body in the revised version to improve clarity.
>
> **Q4. Could you characterize problems where your approach would be inferior to existing methods? E.g., when $N$ is small or when computational resources are limited?**
>
> **A4.**  Thank you for the thoughtful question! As noted in Theorem 13, a smaller number of prediction candidates $N$ actually leads to a tighter bound, and thus does not pose a disadvantage to our method. As for limited computational resources, while our loss involves slightly more structure, as discussed in A3, we have analyzed the time complexity of our loss and found it to be comparable to standard losses such as cross-entropy, both in theory and in terms of empirical runtime. Therefore, even in resource-constrained settings, we do not anticipate a clear disadvantage compared with existing losses.
>
> A limitation may arise when we are interested in second-order optimization. While efficient first-order gradient oracle is provided in Lemma 12 and experimentally validated, the Hessian computation is not very straightforward, which is an interesting direction for future exploration.
>
> [1]. Deng, J., Dong, W., Socher, R., Li, L.-J., Li, K., and Fei-Fei, L. Imagenet: A large-scale hierarchical image database. In CVPR, 2009.
>
> [2]. He, K., Zhang, X., Ren, S., and Sun, J. Deep residual learning for image recognition. In CVPR, 2016.
>
> [3]. Martins, A. and Astudillo, R. From softmax to sparsemax: A sparse model of attention and multi-label classification. In ICML, 2016.
>
> [4]. Charoenphakdee, N., Cui, Z., Zhang, Y., and Sugiyama, M. Classification with rejection based on cost-sensitive classification. In ICML, 2021.
>
> [5]. Cao, Y., Cai, T., Feng, L., Gu, L., Gu, J., An, B., Niu, G., and Sugiyama, M. Generalizing consistent multi-class classification with rejection to be compatible with arbitrary losses. In NeurIPS, 2022.
>
> [6]. Menon, A. K., Rawat, A. S., Reddi, S., and Kumar, S. Multilabel reductions: what is my loss optimising? In NeurIPS, 2019.
>
> [7]. Gao, W., and Zhou, Z. H. On the consistency of multi-label learning. In COLT, 2011.

---

> > ### Comment · Reviewer_jSoZ · 2025-08-04
> >
> > Thank you for addressing my concerns, while the performance improvement lies much below the typical standard deviation for this type of experiment at that level of resulting accuracy, I appreciate the addition.
> > I update my score accordingly.

---

> > > ### Author Response · Authors · 2025-08-05
> > >
> > > Thank you for your reply and update! In addition to the mean accuracy, we observe that the proposed method demonstrates statistical significance over the cross-entropy loss under a 5% t-test, with standard deviations of 0.11% and 0.12% for the two methods, respectively. We appreciate your constructive feedback throughout the review process and are glad that the additional results helped address your concerns.

---

### Official Review · Reviewer_iy2f · 2025-07-13

**Clarity:** 3
**Significance:** 3
**Originality:** 3
**Rating:** 4
**Confidence:** 2

**Summary:**

Previous work has shown that while convex and smooth surrogate losses are effective for estimation and optimization, they often face a trade-off between smoothness and achieving linear regret bounds. This paper overcomes that trade-off by introducing a novel surrogate loss that is both convex and smooth, yet still admits linear surrogate regret bounds. A convolutional negentropy is constructed by encoding the structure of the target loss into a chosen base negentropy, resulting in a convolutional Fenchel–Young loss that inherits both convexity and smoothness. Crucially, the derived prediction link, when paired with this loss, enables a linear surrogate regret bound and provides a consistent probability estimator.

**Questions:**

1. While the theoretical contributions are compelling, I believe the paper’s strengths would be significantly enhanced with empirical validation. For instance, the authors could evaluate the proposed surrogate loss on toy datasets or simple multiclass classification benchmarks with the algorithm presented in Section 4. This would help illustrate the practical impact of the loss and clarify its advantages over existing approaches.

2. I found the definition and role of the term $T(p)$ in Equation (8) somewhat confusing. In line 69, the authors state: *“We encode the structure of a target loss $\ell$ into the chosen base negentropy by adding the negative Bayes risk of $\ell$”*, and in lines 210–211, they mention: *“This polyhedral convex function $T$ is an affinely transformed Bayes risk of a target loss $\ell$”*. While Equation (9) shows that $T $ is an affinely transformed Bayes risk when $p = \mathbb{E}_{y \sim \eta}[\rho(y)] $, this does not seem to align with Equation (11), where $ p = \rho(y) $. Could the authors clarify this discrepancy and more precisely define the conditions under which the interpretation of $ T(p) $ holds?

3. Minor points:
   - Lines 256 and 297: The term *“Theorem”* should be replaced with *“Corollary”*.
   - Line 282: “The regret bound (13)” should be rewritten as “The regret bound in Theorem (13)” or “The regret bound in Eq. (14)”.
   - Equation (18): The notation $ \theta_i $ should be corrected to $\theta_y$.

**Ethical Concerns:**

["NO or VERY MINOR ethics concerns only"]

**Final Justification:**

Before the rebuttal, my main concerns were the lack of empirical results and the unclear definition and role of the term $T(p)$. During the rebuttal, the authors provided additional empirical results on ImageNet and clarified the interpretation of $T(p)$. While these responses addressed my concerns, I am not very familiar with this field and have not carefully reviewed their proof. Therefore, I maintain my score with low confidence.

**Limitations:**

Please see the above questions.

**Quality:**

3

**Strengths And Weaknesses:**

Strengths:

- Proposes a novel surrogate loss that is both convex and smooth while preserving linear surrogate regret bounds.

- The writing and presentation are clear and well-structured.

Weaknesses:

- The paper would be stronger with the inclusion of empirical results to support the theoretical claims.

---

> ### Author Rebuttal · Authors · 2025-07-29
>
> Many thanks for your valuable suggestions! We will fix the typos in the revised manuscript, and the questions and corresponding answers are summarized below:
>
> **Q1. It is beneficial to evaluate the proposed surrogate loss on multiclass classification benchmarks.**
>
> **A1.** Thank you for your advice! We evaluate the performance and report the validation accuracy of our proposed loss in Section 4 on the ImageNet-1k dataset [1] with ResNet-50 [2].
> Our experimental setup follows the default configuration of the PyTorch ImageNet training script:
> we use SGD with a momentum of 0.9, training for 120 epochs with a mini-batch size of 256.
> The initial learning rate is set to 0.1 and divided by 10 every 30 epochs.
> For comparison, we also report results using the standard cross-entropy loss under the same configuration.
> All experiments are conducted using 8 GeForce RTX 4090 GPUs, and we report the average validation accuracy and running time per epoch over 5 runs.
>
> || Cross-entropy Loss | Proposed Loss in Section 4 |
> | ------------------------------| ------------------------------ | -------------------------------------- |
> |Accuracy| 76.45%                         | 76.82%                             |
> |Averaged Running time/Epoch| 643.41 sec |  649.84 sec|
>
> Interestingly, our proposed loss slightly outperforms the standard cross-entropy loss in validation accuracy, as shown in the table above.
> This improvement is observed without any additional tuning, indicating the potential of our approach.
>
>
> **Q2. The interpretation that $T(p)$ is an affinely transformed Bayes risk of the target loss should be defined more precisely.**
>
> **A2.** Thank you for pointing out this issue! We will emphasize in the revised manuscript that this interpretation holds in the case that $p$ is in the convex hull of $\{\rho(1),\cdots,\rho(K)\}$, i.e., there exists $\eta\in\Delta^{K}$ that $p=\mathbb{E}{\small y\sim\eta}[\rho(y)]$. This interpretation also holds when $p=\rho(y)$,which is a special case of $p=\mathbb{E}{\small y\sim\eta}[\rho(y)]$ by setting $\eta =e_{y}.$
>
>
> [1]. Deng, J., Dong, W., Socher, R., Li, L.-J., Li, K., and Fei-Fei, L. Imagenet: A large-scale hierarchical image database. In CVPR, 2009.
>
> [2]. He, K., Zhang, X., Ren, S. and Sun, J. Deep residual learning for image recognition. In CVPR, 2016.

---

> > ### Comment · Reviewer_iy2f · 2025-08-04
> > **Rebuttal response**
> >
> > Thank you for your detailed responses and additional empirical results. While the proposed method has a negligible impact on running time, it offers only a slight improvement in accuracy compared to the standard cross-entropy loss. Besides the average validation accuracy, could the authors also report the variance in accuracy?

---

> > > ### Author Response · Authors · 2025-08-05
> > >
> > > Thank you for your reply! The proposed loss and the cross-entropy loss have standard deviations of 0.11% and 0.12%, respectively, and the improvement is statistically significant under a 5% t-test.
> > > It is rather appealing to observe the statistically significant benefit because the performance improvement on the ImageNet benchmark is substantially challenging and almost saturated.
> > > This evidence supports the convolutional Fenchel-Young loss beyond its theoretical advantages.

---

> > > > ### Comment · Reviewer_iy2f · 2025-08-06
> > > > **Rebuttal response**
> > > >
> > > > Thank you for further clarification about the standard deviations. My concerns have been adequately addressed. I will maintain my score leaning toward acceptance.

---

> > > > > ### Author Response · Authors · 2025-08-06
> > > > >
> > > > > Thank you for your response! We truly appreciate the thoughtful feedback and the time you dedicated to reviewing our work, and we are glad that the additional results helped address your concerns.

---

### Decision · Program_Chairs · 2025-09-17

**Decision:**

Accept (spotlight)

**Comment:**

I have read the paper and really enjoyed it. In the camera-ready version of the paper, I strongly suggest the following:

1- naming. It is fine to use the terminology introduced by Blondel et al but those loss functions have a longstanding history in information geometry which says a lot more than the motivation behind the "Fenchel-Young" naming (which is the remainder of FY inequality). In particular, it is crucial to recognize Shun-ichi Amari's contribution: any FY loss is a Bregman divergence in disguise and the introductory notation is in Amari and Nagaoka's textbook (Methods of Information Geometry), Chapter 3 section 3.4, the canonical divergence (it is exactly eq. 3.44). This says exactly how each argument can be named. Then, there is Leonard J. Savage: FY losses encode the regret of a proper loss, see reference JMLR:v17:14-294 below (proposition 7 point 4). This gives the normative "why" for using such a FY loss (independently of the surrogate regret). I am confident that a nice inclusion of these points in the paper will address the "why such losses" asked by reviewers (e.g. b4uG, DQwU), perhaps by adding a section 2.4 to the paper.

2- notations. Please fix the suggestions of reviewers and make a long read of the paper -- probably by a non-author -- to spot and fix notations. The importance of this step should not be underestimated: it is painful to read a nice paper and to be stopped at times because of imprecise or confusing notations. Example: Algorithm 1 uses notation $[K]$ with two different meanings (index and set). This must be explained and fixed.

3- (no) experiments. I appreciate the effort made by the authors to deliver experiments but I think they are not necessary for the camera-ready (if the authors want, they can eventually push them to the appendix).

@article{JMLR:v17:14-294,
  author  = {Robert C. Williamson and Elodie Vernet and Mark D. Reid},
  title   = {Composite Multiclass Losses},
  journal = {Journal of Machine Learning Research},
  year    = {2016},
  volume  = {17},
  number  = {222},
  pages   = {1--52},
  url     = {http://jmlr.org/papers/v17/14-294.html}
}